# SAMPLING FROM DISCRETE ENERGY-BASED MODELS WITH QUALITY/EFFICIENCY TRADE-OFFS

## ABSTRACT

Energy-Based Models (EBMs) allow for extremely flexible specifications of probability distributions. However, they do not provide a mechanism for obtaining exact samples from these distributions. Monte Carlo techniques can aid us in obtaining samples if some proposal distribution that we can easily sample from is available. For instance, rejection sampling can provide exact samples but is often difficult or impossible to apply due to the need to find a proposal distribution that upper-bounds the target distribution everywhere. Approximate Markov chain Monte Carlo sampling techniques like Metropolis-Hastings are usually easier to design, exploiting a local proposal distribution that performs local edits on an evolving sample. However, these techniques can be inefficient due to the local nature of the proposal distribution and do not provide an estimate of the quality of their samples. In this work, we propose a new approximate sampling technique, Quasi Rejection Sampling (QRS), that allows for a trade-off between sampling efficiency and sampling quality, while providing explicit convergence bounds and diagnostics. QRS capitalizes on the availability of high-quality global proposal distributions obtained from deep learning models. We demonstrate the effectiveness of QRS sampling for discrete EBMs over text for the tasks of controlled text generation with distributional constraints and paraphrase generation. We show that we can sample from such EBMs with arbitrary precision at the cost of sampling efficiency.

## 1 INTRODUCTION

Obtaining samples from a probability distribution is useful in many natural language processing applications, e.g. generating diverse outputs from a language model (Holtzman et al., 2020), producing debiased sentences from a pretrained model (Khalifa et al., 2021), or proposing a set of choices to some decision rule (Eikema & Aziz, 2021). In many cases, when the probability distribution is conveniently factorized and normalized (e.g., autoregressive sequence models), this can be done easily. However, this simplicity in sampling often comes at the cost of expressivity. Energy-based models (EBMs) (LeCun et al., 2006)

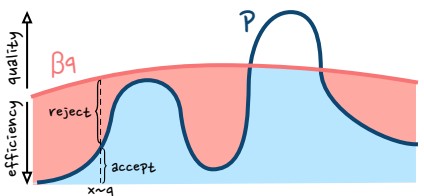

Figure 1: **QRS**. $\beta$ is a scalar parameter that controls the quality/efficiency trade-off. The blue shaded area depicts the truncated distribution that QRS samples from.

enjoy a greater representational freedom by mapping elements into arbitrary unnormalized non-negative scores. In particular, discrete EBMs have been leveraged by prior studies to tackle a number of different tasks, including data-efficient learning (Parshakova et al., 2019a), language modelling (Deng et al., 2020), machine translation (Naskar et al., 2020), natural language understanding (He et al., 2021), and controlled text generation (Khalifa et al., 2021). However, it is not immediately clear how to obtain samples from such EBMs.

In this work we investigate the usage of Monte Carlo (MC) sampling techniques (Robert & Casella, 2004) to approximate the target distribution. MC techniques allow to estimate expectations and to sample from unnormalized distributions, of which discrete EBMs are a subclass, as long as a *proposal* distribution from which it is possible to obtain samples is available. One can distinguish between two kinds of proposal distributions: *local* ones, that is, conditional distributions approximating the normalized target distribution around a given point (for instance, by focusing on a set of local

edits), and *global* unconditional ones that approximate the target distribution over the full sampling space. Because local proposals are generally easier to design, Markov chain Monte Carlo (MCMC) techniques that exploit these, such as Metropolis-Hastings (Metropolis et al., 1953; Hastings, 1970), are quite popular. The convergence theorem for Metropolis-Hastings Markov chains (see Fig. 7 in the Appendix) guarantees that *i)* the expectation estimate converges to the true value over a single chain of increasing length, and *ii)* the distribution obtained by repeatedly running a chain of length $n$ and outputting the $n$-th element converges to the target distribution for increasing values of $n$. These important theoretical results, unfortunately, do not come with concrete convergence diagnostics, which presents serious risks in practice, such as believing that the chain has explored the most relevant parts of the space when it has not (Cowles & Carlin, 1996; Roy, 2020).

Fortunately, there are cases where a reasonable *global* proposal is available. While such cases were the exception in the traditional MC literature, the situation is changing with the advent of powerful neural network training techniques allowing for flexible approximations to a wide class of distributions. We will later show three different instances of global proposal distributions obtained in this way. When a global proposal is available, Metropolis-Hastings reduces to independent Metropolis-Hastings (IMH) (Robert & Casella, 2004, §7.4), where proposed moves do not depend on the current point. While this brings some simplifications in the analysis of the chain, by and large, practical convergence diagnostics remain difficult. If the goal is to compute expectations, the much simpler importance sampling algorithm, which often leads to concrete convergence diagnostics (Owen, 2013), is more appropriate and we will heavily use it in the sequel at points where we need to estimate expectations. If the goal is to produce samples, IMH does not produce i.i.d samples and can suffer from high auto-correlation between samples. Appendix E discusses IMH in detail and provides a theoretical and experimental comparison with our proposed algorithm, QRS, which we introduce next.

With a global proposal, if we can find some constant with which we can bound the importance ratio between the target and proposal distributions, we can use rejection sampling to obtain exact samples (Von Neumann, 1951; Martino et al., 2018). However, it is often hard or impossible to find such a bound. Moreover, if the bound is too large, then the method can be extremely inefficient (Andrieu et al., 2003). In this paper we introduce quasi rejection sampling (QRS), extending rejection sampling by providing the possibility to *approximately* sample from the target distribution without requiring to know a bound on the importance ratio or even that such a bound exists at all. QRS produces i.i.d samples and allows controlling the trade-off between the approximation quality of the sampler and its efficiency. Furthermore, QRS *does* provide explicit convergence diagnostics, while also supplying precise estimates of the sampling quality (i.e., the distance of the sampling distribution to the target distribution) and efficiency (i.e., the sampler's acceptance rate).

We demonstrate the effectiveness of QRS on controlled text generation following the setting of Khalifa et al. (2021) and paraphrase generation inspired by the work of Miao et al. (2019). We sample from EBMs that *i)* restrict a GPT-2 small (Radford et al., 2019) model to generate sequences containing the term "Wikileaks", *ii)* debias GPT-2 fine-tuned on biographies to produce 50% female biographies while exclusively generating biographies about scientists, and *iii)* generate paraphrases that maintain the fluency of GPT-2 while being similar to the input sequence under a sentence embedding model. In order to apply QRS we explore a variety of ways to construct proposal distributions by either *i)* prompting a pre-trained language model, *ii)* training an auto-regressive sequence model to approximate the EBM (Khalifa et al., 2021), or *iii)* making use of off-the-shelf machine translation models to specify conditional proposal distributions. The results show that we are able to approximate the target distributions to arbitrary precision at the cost of sampling efficiency. The trade-off between these two conflicting objectives can be computed explicitly within QRS, allowing user-defined choices depending on the intended application. In short, our contributions are:

- We introduce QRS, a variant of rejection sampling that does not require global upper-bounds on importance ratios, and uses a scalable parameter $\beta$ to trade-off sampling quality with efficiency.

- To support this trade-off, we provide explicit estimates and bounds on discrepancy measures (total variation distance and KL-divergence) between the distribution of QRS samples and the EBM.

- We show how QRS can exploit high quality global proposals readily available today thanks to deep learning. We present experimental results based on three sorts of such proposals, originating from *i)* the use of prompts in pre-trained language models, *ii)* a fine-tuning technique for approximating EBMs, and *iii)* the use of round-trip translation in the context of a paraphrasing EBM.

## 2 FORMAL APPROACH

Our general problem is the following. We consider a discrete (i.e. countable) sample space $X$.[1] We are given a nonnegative real function — aka EBM — $P(x)$ over $X$, such that the partition function $Z \doteq \sum_{x \in X} P(x)$ is strictly positive and finite. We can then associate with $P$ a normalized probability distribution $p(x) \doteq Z^{-1}P(x)$. Our goal is to define a "sampler" $\omega$, that is a generator of elements from $X$, such that $\omega$ produces a sample $x$ with a probability $\omega(x)$ as close as possible to our target $p(x)$, in terms of discrepancy measures such as KL-divergence $D_{\mathrm{KL}}(p, \omega)$ and total variation distance $\mathrm{TVD}(p, \omega)$, to be detailed later. To help us solve that problem, we assume that we have at our disposal a *global proposal* distribution $q(x)$ such that *i)* we can effectively compute $q(x)$ (i.e. *score* $x$) for any $x \in X$, *ii)* we can effectively generate samples from $q$, and *iii)* the support of $q$ includes the support of $p$, i.e. $p(x) > 0 \rightarrow q(x) > 0$ (but see footnote 3 for a generalization).

Additionally, and crucially for the tractability of the techniques, the proposal $q$ should be chosen in such a way that it provides a "reasonable" starting point towards our target $p$, in terms of $\mathrm{TVD}$ or $D_{\mathrm{KL}}$. One important methodological contribution of our approach will be to stress the role of deep learning in producing good proposals in a way that is not typical of classical MCMC approaches.

### 2.1 QUASI REJECTION SAMPLING (QRS)

Our proposed approach, QRS, is based on Algorithm 1.

---
**Algorithm 1** QRS [rejection sampling (RS)]

---
**Require:** $P, q, \beta$                                                         $\triangleright\; 0 < \beta < \infty$
 1: **while** True **do**
 2:      $x \sim q$
 3:      $r_x \leftarrow \min(1, P(x)/\beta q(x))$    [vs.    $r_x \leftarrow P(x)/\beta q(x)$]            $\triangleright$ Acceptance prob.
 4:      $u \sim U_{[0,1]}$                                   $\triangleright\; U_{[0,1]}$ : unif. dist. over $[0, 1]$
 5:      **if** $u \leq r_x$ **then**
 6:          output $x$

---

In addition to $P$ and $q$, QRS requires the input of a finite positive number $\beta$. QRS differs from standard rejection sampling in two aspects: *i)* contrary to rejection sampling it does not require $\beta$ to be a "global" upper-bound, that is, to have $P(x)/q(x) \leq \beta$ for all $x$'s in $X$, and (2), as shown on line 3, the "acceptance probability" $r_x$ is a generalization of the one used with rejection sampling, for cases where $P(x)/\beta q(x) > 1$. In case $\beta$ happens to be a global upper-bound, QRS simplifies to standard rejection sampling. See Fig. 1 for an illustration of QRS.

Both rejection sampling and QRS produce an i.i.d sequence of $x$'s (line 6), where each $x$ is generated with a probability that we will denote as $p_\beta(x)$. As is well-known (Robert & Casella, 2004), in the case of rejection sampling, we actually have $p_\beta = p$. In other words, rejection sampling is a *perfect* sampler for $p$. This is of course a major advantage, however it comes with serious theoretical and practical limits: *i)* rejection sampling requires the existence of a *finite* upper-bound $\beta$, *ii)* this $\beta$ needs to be known beforehand. These conditions often do not hold for the proposals $q$ that we will consider, typically autoregressive models whose statistics are not known in closed form. Even if such a bound could be found, the resulting sampler could be extremely inefficient: as we will see (Equation 2), the "acceptance rate" of rejection sampling is proportional to $1/\beta$, which can be extremely small.

By relaxing the requirement that $\beta$ be a global upper-bound, QRS loses the identity between $p_\beta$ and $p$. However, QRS becomes much more generally applicable, and crucially, allows an explicit trade-off between the sampling *efficiency* of $p_\beta$ and its sampling *quality*, as measured by distributional discrepancy between $p_\beta$ and $p$.[2] Let's now look at this in more detail.

---

[1]In our experiments $X$ will mostly be a set of finite sequences over linguistic tokens, but here we consider an arbitrary discrete space.

[2]We note that for off-the-shelf usage, we provide in Appendix D a specific variant of the algorithm that automatically estimates the best possible $\beta$ given some efficiency constraints.

## 2.2 Formal properties of QRS

We will keep the notation from above, and will also be using two standard discrepancy measures between distributions $p_1, p_2$ over $X$: the KL-divergence $D_{KL}(p_1, p_2) \doteq \mathbb{E}_{x \sim p_1} \log[p_1(x)/p_2(x)]$, and the total variation distance $\mathrm{TVD}(p_1, p_2) \doteq 1/2 \sum_x |p_1(x) - p_2(x)|$ (see e.g. Chafaï (2010)). Both measures are null iff $p_1 = p_2$, with TVD (resp. KL) ranging between 0 and 1 (resp. 0 and $\infty$).

The following facts are proven in Appendix A:

- Define $P_\beta(x) \doteq \min(P(x), \beta q(x))$, and let $Z_\beta \doteq \sum_{x \in X} P_\beta(x)$ be the partition function of $P_\beta(x)$. Then $p_\beta$ is the normalized distribution associated with $P_\beta$, with:

$$p_\beta(x) = 1/Z_\beta \, P_\beta(x). \tag{1}$$

- By definition, the acceptance rate $\mathrm{AR}_\beta$ of the QRS sampler $p_\beta$ is the proportion of samples from $q$, in line 2 of the algorithm, that produce an output on line 6. $\mathrm{AR}_\beta$ is a decreasing function of $\beta$, and:

$$\mathrm{AR}_\beta = \mathbb{E}_{x \sim q} \min(1, P(x)/\beta q(x)) = Z_\beta/\beta. \tag{2}$$

- Let $A_\beta \doteq \{x \in X : P(x)/q(x) \le \beta\}$. Then:

$$\mathrm{TVD}(p, p_\beta) \le 1 - p(A_\beta), \tag{3}$$

$$\lim_{\beta \to \infty} (1 - p(A_\beta)) = 0. \tag{4}$$

In other words, $1 - p(A_\beta)$ (which can be understood as the probability of violating $P(x)/q(x) \le \beta$) bounds the TVD between $p_\beta$ and $p$, and $p_\beta$ converges to $p$ for $\beta \to \infty$.[3]

## 2.3 Practical Implications: Estimates

The previous facts have important practical implications, in particular concerning the production of *explicit* estimates for different quantities of interest. The general recipe for producing such estimates will be to use importance sampling (Owen, 2013), using once again $q$ as the proposal distribution. We base all estimates on a sample $\{x_1, \ldots, x_N\}$ of i.i.d draws from $q$; if $f$ is a real-valued function on $X$, we then rewrite $\sum_{x \in X} f(x) = \mathbb{E}_{x \sim q} \frac{f(x)}{q(x)} \simeq N^{-1} \sum_{i \in [1,N]} \frac{f(x_i)}{q(x_i)}$. In particular, we have:

$$Z \simeq N^{-1} \sum_{i \in [1,N]} \frac{P(x_i)}{q(x_i)}, \quad (5) \qquad p(A_\beta) \simeq N^{-1} \sum_{i \in [1,N]} \frac{P(x_i)}{Zq(x_i)} \mathbb{1}[x_i \in A_\beta], \quad (7)$$

$$Z_\beta \simeq N^{-1} \sum_{i \in [1,N]} \frac{P_\beta(x)}{q(x_i)}, \quad (6) \qquad \mathbb{E}_{x \sim p_\beta} f(x) \simeq N^{-1} \sum_{i \in [1,N]} \frac{P_\beta(x_i)}{Z_\beta q(x_i)} f(x_i), \quad (8)$$

where we note that explicit values for $P_\beta(x)$ and $\mathbb{1}[x_i \in A_\beta]$ are available, namely: $P_\beta(x) \doteq \min(P(x), \beta q(x))$ and $\mathbb{1}[x_i \in A_\beta] = 1$ iff $P(x) \le \beta q(x)$.

We can use these estimates to obtain estimates of the discrepancies between $p$ and $p_\beta$, again by importance sampling with $q$. We have (see Appendix B):

$$\mathrm{TVD}(p, p_\beta) \simeq 1/2 \, N^{-1} \sum_{i \in [1,N]} \left| \frac{P(x_i)}{Zq(x_i)} - \frac{P_\beta(x_i)}{Z_\beta q(x_i)} \right|, \tag{9}$$

$$D_{KL}(p, p_\beta) \simeq \log \frac{Z_\beta}{Z} + N^{-1} \sum_{i \in [1,N]} \frac{P(x_i)}{Zq(x_i)} \log \frac{P(x_i)}{P_\beta(x_i)}. \tag{10}$$

## 3 Experiments

### 3.1 Two Poissons

To demonstrate how we can use the QRS algorithm to obtain samples from a distribution $p(x)$ using a proposal distribution $q(x)$ we start with a toy setting using two Poissons. The goal is to sample

---

[3]In all this discussion, we have kept the standard assumption about supports of $p$ and $q$, namely that $\mathrm{Supp}(p) \subseteq \mathrm{Supp}(q)$. Interestingly, when this assumption is not true, all the previous properties of QRS still hold, apart from (4), which generalizes to $\lim_{\beta \to \infty}(1 - p(A_\beta)) = 1 - p(\mathrm{Supp}(q))$ (see Eq. (18) in App. A).

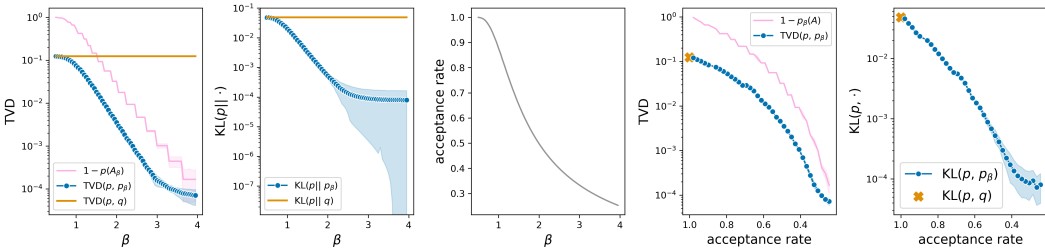

Figure 2: Estimation of sampling quality ($\text{TVD}(p, p_\beta)$ and $D_{\text{KL}}(p, p_\beta)$), efficiency (acceptance rate), and the trade-off between them for a QRS sampler when using a proposal $q = \text{Poisson}(\lambda = 10)$ to approximate $p = \text{Poisson}(\lambda = 11)$ over five independent experiments with 10M samples.

from a target Poisson distribution $p$ with rate $\lambda_p = 11$ using samples from a proposal Poisson distribution $q$ with rate $\lambda_q = 10$. Rejection sampling is not possible in this setting as the ratio $p(x)/q(x) = (e^{-11}11^x/x!)/(e^{-10}10^x/x!) = e^{-1} 1.1^x$ can take arbitrarily large values when $x$ increases, i.e. is unbounded. However, it is still possible and practical to use the QRS sampler.

We perform five independent experiments in which we sample 10M elements from $q$ that we use to compute the quality of the approximation by estimating: $\text{TVD}(p, p_\beta)$ (Eq. 9), its upper bound $1 - p(A_\beta)$ (Eq.3), and $D_{\text{KL}}(p, p_\beta)$ (Eq. 10) measured in nats/sequence. In all cases, we use a range of $\beta$ values in the interval $[0.5, 4]$. Furthermore, we compute the sampler's efficiency by estimating the acceptance rate (AR) for each value of $\beta$ following Equations 2 and 6. To visualize the trade-off between quality and efficiency, we also plot one as a function of the other by computing the inverse of the AR curve and composing it with the TVD and KL ones.

We first display quality and efficiency results as a function of $\beta$ (first three panels in Fig. 2). As shown, using higher values of $\beta$ improves the TVD and KL, even though this comes at the cost of lower acceptance rate. The last two panels in Fig. 2 show the quality/efficiency trade-off in a more concise form, which is why we will prefer this presentation in experiments below. As shown, the TVD reaches very low values ($< 10^{-4}$) with a moderate acceptance rate of $0.25$.

## 3.2 Generation with Distributional Control

Our following experiments focus on the generation with distributional control setting introduced by Khalifa et al. (2021). Given a language model $a(x)$, the goal of this task is to obtain a model $p(x)$ that, on the one hand, constrains the moments of a set of $n$ pre-defined features $\phi(x)$ to match some desired values $\bar{\mu}$ (i.e., $\mathbb{E}_{x \sim p}\phi(x) = \bar{\mu}$), while on the other hand minimizing $D_{\text{KL}}(p, a)$. For example, one might want to debias a language model trained on a corpus of biographies to produce biographies only of scientists, 50% of which should be female. Then $\phi_1(x)$ and $\phi_2(x)$ would be binary classifiers assessing whether a sentence speaks about a scientist or female person respectively, and the desired moments would be set to $\bar{\mu} = [1, 0.5]$.

The authors show that $p$ can be expressed as an unnormalized EBM $P(x) = a(x)b(x)$, and describe two variations. On the one hand, they consider *pointwise* constraints, where $\bar{\mu} \in \{0, 1\}^n$. For instance, if there is a single binary feature for which we would like that $\forall x : \phi(x) = 1$, then $b$ takes the form $b(x) = \phi(x)$. Otherwise, in the case of *distributional* constraints in which $\bar{\mu} \in \mathbb{R}^n$, they show that there is a vector $\lambda \in \mathbb{R}^n$ such that $b(x) = \exp(\lambda \cdot \phi(x))^4$ such that $p(x) \propto a(x)b(x)$ fulfills the requirements of moment matching and minimal KL distance from the original model. Finding this vector of $\lambda$ parameters is done through a combination of self-normalized importance sampling (Owen, 2013; Parshakova et al., 2019a) and stochastic optimization.

### 3.2.1 Proposal distributions for a pointwise constraint

We first experiment with constraining GPT-2 small (Radford et al., 2019) using one of the pointwise constraints ($\bar{\mu} = 1.0$) proposed in Khalifa et al. (2021), namely, $b(x) = \mathbb{1}[x$ contains "Wikileaks"$]$. In order to apply QRS we need to find a suitable proposal distribution. A possible candidate is

---

[4]For a precise formulation covering all cases, see (Khalifa et al., 2021).

| prompt-name | prompt |
|---|---|
| simple | Wikileaks. |
| multiple | Wikileaks, Wikileaks, Wikileaks. |
| knowledge | Here is what I know about Wikileaks: |
| jeopardy | This medium was founded by Julian Assange in 2006. |
| news | Here are the latest developments on Wikileaks: |

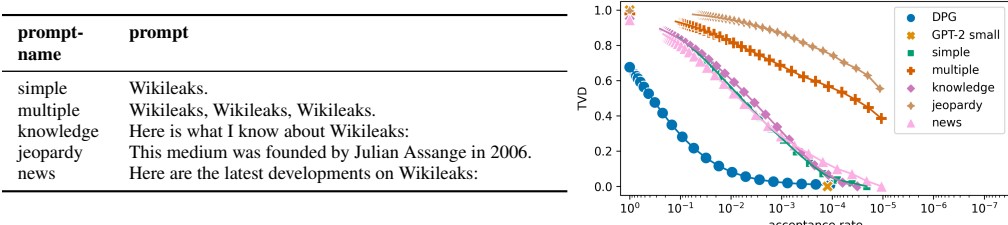

Figure 3: GPT-2 generations containing the term "Wikileaks". We use GPT-2 small, GPT-2 small conditioned on various prompts and the fine-tuned model obtained through DPG (Khalifa et al., 2021) as proposal distributions $q$ for the QRS algorithm. Points in the left-upper corner correspond to $\mathrm{TVD}(p, q)$ before QRS, while the curves show $\mathrm{TVD}(p, p_\beta)$ as a function of the acceptance rate.

GPT-2 small itself. An advantage of this proposal is that we can use pure rejection sampling with an upper-bound $\beta = 1$ to obtain exact samples from the EBM. This is because we can upper bound the ratio $P(x)/q(x) = a(x)b(x)/a(x) = b(x) \leq 1$. Furthermore, for $b(x) \in \{0, 1\}$ this process reduces to "naively" filtering out all samples for which $b(x) = 0$. However, a serious disadvantage is that the acceptance rate will be given by the natural frequency of the constraint (in this case, in the order of $10^{-4}$). Using QRS, we can employ proposal distributions leading to better efficiency at a small cost in quality of approximation to $p$. We explore two such options. First, we make use of the model proposed by Khalifa et al. (2021), which consists of a fine-tuned auto-regressive model obtained from applying the distributional policy gradient (DPG) algorithm (Parshakova et al., 2019a) to approximate the target EBM. While this model is considerably better at satisfying the desired constraints, it does not match the desired distribution perfectly. Second, we propose to condition $a(x)$ on a prompt with the aim to increase the constraint satisfaction rate in the resulting conditional distributions. In contrast to the previous approach, this solution does not require training a model, even though it does require to find the prompts themselves. We experiment with five such prompts, which we present in Fig. 3.

The right panel of Fig. 3 shows the $\mathrm{TVD}(p, \cdot)$ as a function of the acceptance rate for different samplers. For this and following experiments, we chose a range of $\beta$ values that yield acceptance rates in a range $10^0$–$10^{-5}$ (cf. Table 1 in Appendix). We first show the $\mathrm{TVD}(p, q)$ for each proposal $q$, at an acceptance rate of 1, before applying QRS. Then, we plot $\mathrm{TVD}(p, p_\beta)$ as a function of acceptance rate, running the QRS sampler for each of the various proposal distributions. We compute importance sampling estimates of the TVD on 1M samples from each proposal distribution. As expected, using GPT-2 small comes with perfect TVD at the cost of low efficiency with an acceptance rate around $10^{-4}$. Using prompting, we can improve the constraint satisfaction of the resulting proposal distributions and trade-off quality for efficiency using QRS. Some prompts work notably better than others and we do not exclude the possibility of there existing prompts that perform even better than the ones we tested. We leave a more extensive exploration of prompting to create proposal distributions to future work. The auto-regressive policy obtained from the DPG algorithm is the best proposal distribution we tested. Notably, it allows for obtaining very low TVD values at a higher acceptance rate than would be obtained by naively filtering samples from the base language model.

### 3.2.2 DEBIASED SCIENTIST BIOGRAPHIES

We now turn to the task, also introduced by Khalifa et al. (2021), of generating biographies of scientists while debiasing the gender distribution to contain female scientists 50% of the time. For this we make use of GPT-2 Biographies ($a(x)$), a language model fine-tuned on Wikipedia biographies[5] and follow the same the setup as the authors to define the binary classifiers identifying sequences talking about scientists or female figures[6] and infer an EBM that matches the distributional constraints with minimal deviation from the original model. The frequency of $a(x)$ generating scientist biographies is 1.8%, female biographies 7.5%, and female scientist biographies only 0.14%. As proposal distribution we use the DPG model that Khalifa et al. (2021) trained to approximate the EBM, which reaches a constraint satisfaction of 69.0% scientist, 27.3% female and 19.6% female scientist biographies.

---

[5] https://huggingface.co/mkhalifa/gpt2-biographies

[6] Gender is estimated by the ratio of female to male pronoun counts, scientists are identified by the mention of at least one of multiple words associated with the profession.

**QRS samples** from $p$ at AR $= 10^{-3}$

Chandra Pradha Towni (born February 11, 1965) is a social scientist, activist, poet, and author living in Portugal. She is. . .

Enrella Carrière is a Canadian writer, translator, and philosopher specializing in the history of show business. She has covered topics such as the direction and psychology of television and the evolution of human. . .

Albert Fahn (born 1970) is an American scientist who focuses on algorithms for generating biomechanical data. Methods to generate and construct biomechanical data from. . .

Wyndham Radnor (born 1946) is a British historian and criminologist specialising in the subject of labour law. He has written extensively on. . .

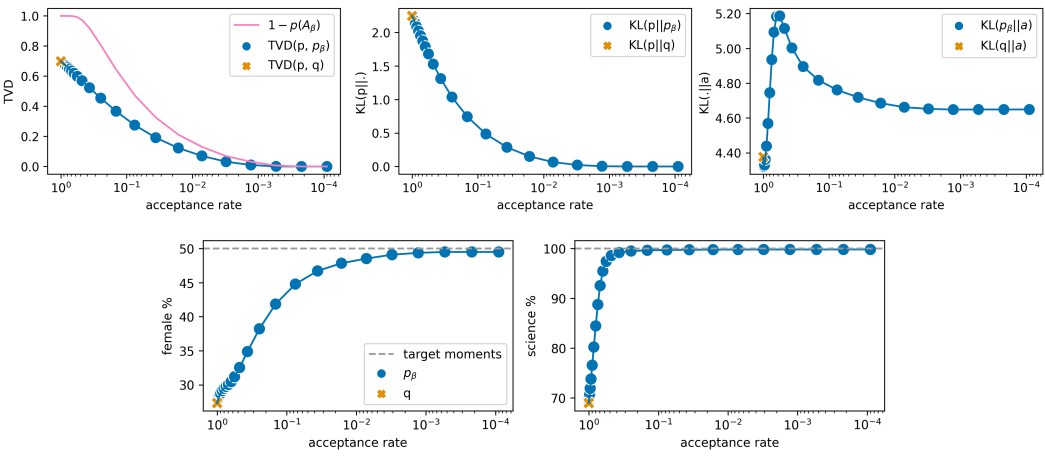

Figure 4: Estimation of the divergence from the EBM (TVD and KL to $p$), moments of features *female* and *science*, and divergence from the original base model (KL from $a$) to debias GPT-2 biographies talking about scientists. We also show samples from running the QRS sampler at $10^{-3}$ acceptance rate. Samples are cut off at 40 (subword) tokens and are manually chosen to show two male and two female biographies, for constraint satisfaction (moment matching) results refer to the graph. We color words that fire our female or science features.

As before, we obtain 1M samples from the proposal distribution to compute importance sampling estimates of quality and efficiency metrics (i.e., $\mathrm{TVD}(p, p_\beta)$, $D_{\mathrm{KL}}(p||p_\beta)$ and AR), plus the backward KL-divergence from the base language model $D_{\mathrm{KL}}(\cdot||a)$ (Eq. 27, in Appendix B) and the moments of the features that we wish to control (Eq. 8). We show all metric curves as a function of acceptance rate of the QRS algorithm as well as some example generations in Fig. 4.

We find that $\mathrm{TVD}(p, p_\beta)$ and $D_{\mathrm{KL}}(p||p_\beta)$ (as well as the upper-bound on $\mathrm{TVD}(p, p_\beta)$) steadily converge to 0 as the acceptance rate decreases, meaning that we can perfectly match the target EBM at the cost of sampling efficiency. As a result, at a moderate acceptance rate AR $= 10^{-3}$ we nearly perfectly debias our original language model while exclusively having it generate biographies about scientists (49.5% female and 99.8% scientist biographies). We show some example generations at AR $= 10^{-3}$ chosen manually to show two male and two female biographies. Notably, we also achieve very decent constraint satisfaction (44.8% female and 99.7% scientist biographies) and TVD (at 0.27) already at AR $= 10^{-1}$, allowing to considerably improve the quality at a small cost in efficiency. Divergence to the original language model $D_{\mathrm{KL}}(p_\beta||a)$ steeply increases as the feature moments are matched more closely, after which it gradually decreases before stabilizing. This reflects the construction of the EBM, which projects $a(x)$ onto the constraint manifold in such a way that the KL-divergence from the original language model is minimized. Two more pointwise and two more distributional constraints are shown in Appendix G with similar results.

### 3.3 PARAPHRASING

Finally, inspired by Miao et al. (2019), we do proof-of-concept experiments on paraphrase generation by framing it as a conditional EBM. Specifically, given a sentence $y$ to paraphrase, we define our EBM through a pointwise constraint on $a(x) = \text{GPT-2}(x)$, with $b(x)$ a binary classifier that classifies a pair $(x, y)$ as a paraphrase if the cosine similarity between their sentence embeddings is above 0.95.

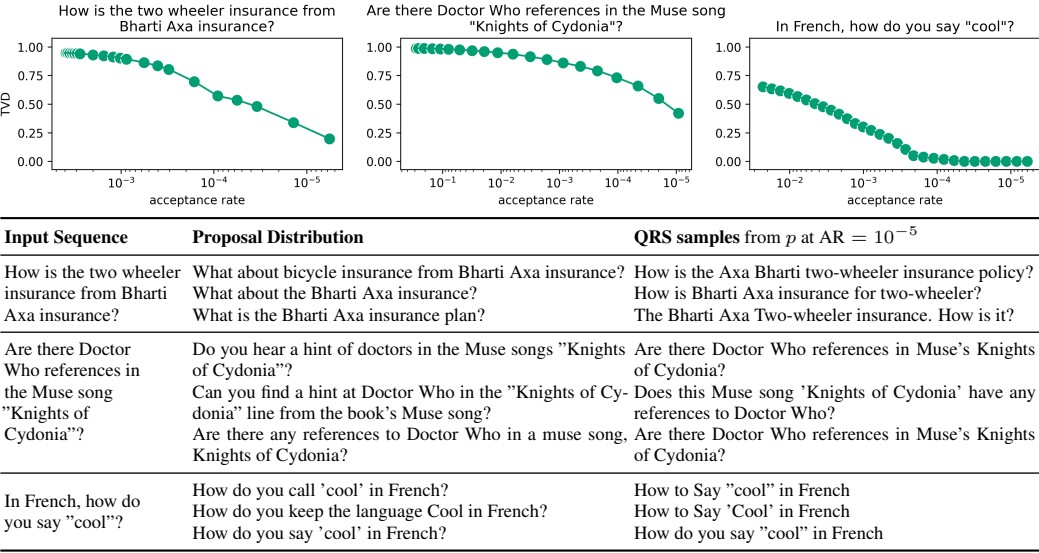

| Input Sequence | Proposal Distribution | QRS samples from $p$ at AR $= 10^{-5}$ |
|---|---|---|
| How is the two wheeler insurance from Bharti Axa insurance? | What about bicycle insurance from Bharti Axa insurance? What about the Bharti Axa insurance? What is the Bharti Axa insurance plan? | How is the Axa Bharti two-wheeler insurance policy? How is Bharti Axa insurance for two-wheeler? The Bharti Axa Two-wheeler insurance. How is it? |
| Are there Doctor Who references in the Muse song "Knights of Cydonia"? | Do you hear a hint of doctors in the Muse songs "Knights of Cydonia"? Can you find a hint at Doctor Who in the "Knights of Cydonia" line from the book's Muse song? Are there any references to Doctor Who in a muse song, Knights of Cydonia? | Are there Doctor Who references in Muse's Knights of Cydonia? Does this Muse song 'Knights of Cydonia' have any references to Doctor Who? Are there Doctor Who references in Muse's Knights of Cydonia? |
| In French, how do you say "cool"? | How do you call 'cool' in French? How do you keep the language Cool in French? How do you say 'cool' in French? | How to Say "cool" in French How to Say 'Cool' in French How do you say "cool" in French |

Figure 5: $\text{TVD}(p, p_\beta)$ running the QRS sampler at various acceptance rates to generate paraphrases of three sequences (**top**). We show some example paraphrases from both the proposal distribution $q(x)$ (round-trip NMT) as well as the QRS sampler $p_\beta$ at an acceptance rate of $10^{-5}$ (**bottom**).

We obtain high-quality sentence embeddings from sentence BERT[7] (Reimers & Gurevych, 2019). As proposal distribution we will not be using GPT-2, but rather illustrate how we can utilize off-the-shelf deep learning models as proposal distributions for QRS. In particular, we use a round-trip MT model, which is a well-known tool in generating paraphrases (Bannard & Callison-Burch, 2005; Mallinson et al., 2017). Specifically, we use the English-to-German and German-to-English models from Ng et al. (2019). We first obtain a beam searched (Graves, 2012) translation into German,[8] and then define the proposal distribution as the German-to-English model conditioned on the beam searched translation. We locally renormalize the model to do top-30 sampling (Fan et al., 2018).

We show importance sampling estimates of $\text{TVD}(p, p_\beta)$ using 1M samples for three sequences in Fig. 5 along with example samples from both the proposal distribution and QRS at AR $= 10^{-5}$. The proposal distribution quality varies per input sequence as can be seen by the slope of the curve and the low-efficiency starting points of some curves (non-paraphrases are always rejected and so have a big influence on the acceptance rate). Still, QRS allows us to approximate the target EBM reasonably well at the cost of sampling efficiency ($\text{TVD}(p, p_\beta)$ is $0.20$, $0.42$ and near $0$). Looking at the examples, we find the proposal distribution to produce decent paraphrases, but not always semantically equivalent or grammatically correct. The QRS samples are mostly semantically equivalent, though they still produce some mistakes ("Axa Bharthi" vs "Bharti Axa") and seem to be insensitive to the question mark and to the casing of words ("Cool", "Two-wheeler insurance"). Interestingly, this experiment illustrates how the presented approach could be employed to *disentangle* the questions of how to model a problem (by defining the corresponding EBM) and how to efficiently sample from it (by improving the proposal distributions), allowing to work on each of these questions separately.

## 4 RELATED WORK[9]

Sampling from discrete EBMs does not require an MC approach. DPG (Parshakova et al., 2019b) instead trains an autoregressive model that approximates a given EBM, which can then be used to obtain samples efficiently. However, it can be hard or impossible, in practice, to approximate the target EBM accurately because of its complexity, or more fundamentally, because of inherent

---

[7]We use `https://huggingface.co/sentence-transformers/all-mpnet-base-v2`.

[8]We use a beam size of $5$.

[9]For more related work, in particular about continuous EBMs, and also the use of sampling techniques in natural language processing, please see App. F.

limitations of autoregressive architectures (Lin et al., 2021). For instance, while empirical results by Khalifa et al. (2021) show that their DPG-based approximation is reasonable, there is still a sizeable gap to the target distribution. Here, we close the gap almost completely at moderate cost in sampling efficiency by using the approximation as a proposal distribution for QRS. For this paper we were inspired by the ability of some Monte Carlo approaches to guarantee accurate sampling in the limit. Without trying to cover the full landscape here, we provide some pointers to relevant MC techniques that can exploit global proposal distributions. Possibly the most obvious MCMC candidate, as already mentioned, is IMH (Robert & Casella, 2004, §7.4); we devote Appendix E to a detailed comparison of IMH with QRS. Due to the non i.i.d. nature of the associated sampler, to its inability to score its samples, and the resulting unavailability of explicit convergence diagnostics, we then argue for the superiority of QRS for our purposes. Several other approaches than ours have taken rejection sampling as their starting point. Similar to us, Rejection Sampling Chains (Tierney, 1992), (Chib & Greenberg, 1995, §6.1) do not require a global upper-bound. It is a hybrid that uses rejection sampling in a region satisfying a partial upper-bound but combines it with IMH outside of that region to produce a Markov chain that converges to the correct stationary distribution in the limit. Caffo et al. (2002) propose Empirical Supremum Rejection Sampling, an algorithm that adaptively increases the $\beta$ upper-bound based on the maximum observed so far, with a focus on convergence in the limit rather than approximation quality. Some researchers have observed before us that a partial bound $\beta$ leads to the probability distribution presented in Eq. 1. Rejection Control (Liu et al., 1998), (Liu, 2004, pp. 44-45) makes use of this observation for accelerating the computation of an unbiased importance sampling estimate of the expectation $\mathbb{E}_{x \sim p} f(x)$ in situations where computing $f(x)$ is expensive and where it is desired to reduce such computation in regions of $p$ for which the importance ratios are small and have less impact on the importance sampling estimate. None of the above approaches provide explicit convergence diagnostics nor consider a trade-off between efficiency and approximation accuracy. We argue that such a trade-off is important for practical use-cases of sampling, where fast response time may be prioritized over obtaining exact samples.

## 5 CONCLUSIONS

In this paper, we address the problem of obtaining samples from a target EBM given access to a good global proposal distribution. In general, a technique that addresses this question has the potential to *decouple*, for any generative task, modelling (by tuning the EBM definition) from efficient sampling (by tuning the proposal distribution). In the past, high-quality global proposals were typically not easy to come by, thus strongly motivating the development of MCMC techniques which could exploit simple local proposals that computed transition probabilities between candidate samples. Today, however, developments in neural network training techniques make high-quality global proposal distributions easier to obtain than before. Motivated by such developments we propose QRS, a generalization of rejection sampling that exploits such proposals to approximately sample from an EBM. QRS can be applied even in cases in which no upper bound of the importance ratio between the EBM and the proposal distribution scores is known, or even exists at all. Notably, QRS also provides strong theoretical guarantees, which include not only convergence to the target distribution (Eq. 4), but also diagnostics that are not available for other MCMC sampling techniques like independent Metropolis-Hastings, such as an upper bound on the TVD (Eq. 3) or unbiased estimators of the TVD and KL divergence to the target distribution (Eqs. 9 - 10). Our experimental results on discrete EBMs show that QRS achieves strong results on the studied controlled text generation setting where, for instance, the sampler achieves excellent debiasing of the language model using acceptance rates in the range of $10^{-1}$ to $10^{-3}$. Furthermore, we show the versatility of the approach by showing how it can be applied to sample paraphrases from an EBM formulation derived from combining a pretrained language model and a sentence similarity score. Yet, there is a trade-off between quality and efficiency, and finding the right balance will depend on the particular application. QRS allows to use any arbitrary value of $\beta$, producing levels of quality and efficiency that can be estimated. For convenience, in Appendix D we also describe a variant of QRS that automatically adjusts $\beta$ under the constraint of not falling below a target acceptance rate. Last, we note that while here we have focused on EBMs for discrete sequential spaces, nothing prevents QRS from being applied to continuous spaces, making it potentially useful for such applications as speech or vision.

## REPRODUCIBILITY STATEMENT

To ensure the full reproducibility of this work, we have included in Appendix A and B complete proofs and derivations of all propositions and quantities in this paper. Furthermore, source code of all experiments is provided as an anonymized link to the reviewers and will be open-sourced upon publication of this manuscript. Lastly, a table with the range of $\beta$ values used to generate the plots is given in Table 1 in the Appendix.

## ETHICS STATEMENT

Part of the experimental section of this paper studies the generation with distributional control task, defined by Khalifa et al. (2021). The goal of this task is to adapt the generations of a language model to a given set of norms that should be quantifiable as preferences over the moments of certain features. This formalization has the potential of addressing numerous problems related to social bias in large pretrained language models (Sheng et al., 2019; Liang et al., 2021), including gender bias. Nonetheless, we note that the approach is not prescriptive with respect to which norms should be applied, nor about how to quantify the relevant features, all of which can be decided with the relevant stakeholders. Finally, we note that all experiments have been done on the English language due to the availability of large pre-trained language models in this language. We believe that the presented set of techniques should be equally applicable to other languages and multilingual models as well.

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

## A   PROPERTIES OF QRS: PROOFS

**Equation (1):** By definition $p_\beta(x)$ is the probability that the first[10] output from Algorithm 1 is equal to $x$. On the first step of the algorithm, the probability that a given $x$ is accepted is $q(x)r_x$, while the probability that no $x$ at all is accepted is $\rho \doteq \sum_{x \in X} q(x)(1 - r_x) = 1 - \sum_{x \in X} q(x)r_x$. More generally, the probability for $x$ to be accepted on step $i$ of the algorithm, while no $x$ was accepted on previous steps is then $\rho^{i-1}q(x)r_x$. Overall, the probability $p_\beta(x)$ of $x$ to be the *first $x$* to be accepted is:

$$\sum_{i=1}^{\infty} \rho^{i-1}q(x)r_x = q(x)r_x \sum_{i=1}^{\infty} \rho^{i-1} = \frac{1}{1-\rho}q(x)r_x \tag{11}$$

$$= \frac{1}{\sum_{x \in X} q(x)r_x}q(x)r_x = \frac{1}{Z_\beta}P_\beta(x). \tag{12}$$

**Equation (2):** We have:

$$\mathrm{AR}_\beta = \mathbb{E}_{x \sim q}\min(1, P(x)/\beta q(x)) = \beta^{-1}\sum_{x \in X}\min(P(x), \beta q(x)) \tag{13}$$

$$= \beta^{-1}\sum_{x \in X}P_\beta(x) = Z_\beta/\beta. \tag{14}$$

**Equation (3):** For the proof, we will need a well-known property of TVD (Chafaï, 2010), namely that, for any distributions $p_1, p_2$ over $X$, we have:

$$\mathrm{TVD}(p_1, p_2) = \sum_{x \in X: p_1(x) \geq p_2(x)} p_1(x) - p_2(x). \tag{15}$$

The reason is simple; we have:

$$\left[\sum_{x \in X: p_1(x) \geq p_2(x)} p_1(x) - p_2(x)\right] - \left[\sum_{x \in X: p_1(x) < p_2(x)} p_2(x) - p_1(x)\right]$$

$$= \sum_{x \in X} p_1(x) - \sum_{x \in X} p_2(x) = 0,$$

which proves that the first and second expressions under brackets are equal. Then we have:

$$\mathrm{TVD}(p_1, p_2) = 1/2 \sum_{x \in X} |p_1(x) - p_2(x)|$$

$$= 1/2\left[\sum_{x \in X: p_1(x) \geq p_2(x)} p_1(x) - p_2(x)\right] + 1/2\left[\sum_{x \in X: p_1(x) < p_2(x)} p_2(x) - p_1(x)\right]$$

$$= \sum_{x \in X: p_1(x) \geq p_2(x)} p_1(x) - p_2(x).$$

Our **main proof**, illustrated in Figure 6, proceeds as follows.
Let $A_\beta \doteq \{x \in X : P(x) \leq \beta q(x)\}$ and $\bar{A}_\beta \doteq X \setminus A_\beta$.
We have $P_\beta(x) \doteq \min(P(x), \beta q(x))$ and therefore $P_\beta(x) = P(x)$ for $x \in A_\beta$, and $P_\beta(x) < P(x)$ for $x \in \bar{A}_\beta$. Overall $P_\beta$ is smaller or equal to $P$ and thus $Z_\beta \leq Z$. For any $x$, we have $p_\beta(x) = Z_\beta^{-1}P_\beta(x)$ and $p(x) = Z^{-1}P(x)$, and hence for $x \in A_\beta$, $p(x) \leq p_\beta(x)$.
If we define $C_\beta \doteq \{x \in X : p(x) \leq p_\beta(x)\}$, and $\bar{C}_\beta \doteq X \setminus C_\beta$, we have $A_\beta \subseteq C_\beta$ and $\bar{C}_\beta \subseteq \bar{A}_\beta$.
Using Equation (15), we now see that $\mathrm{TVD}(p, p_\beta) = \sum_{x \in X: p(x) \geq p_\beta(x)} p(x) - p_\beta(x) = \sum_{x \in X: p(x) > p_\beta(x)} p(x) - p_\beta(x) = \sum_{x \in \bar{C}_\beta} p(x) - p_\beta(x) \leq \sum_{x \in \bar{C}_\beta} p(x)$.
Finally we get:

$$\mathrm{TVD}(p, p_\beta) \leq p(\bar{C}_\beta) \leq p(\bar{A}_\beta) = 1 - p(A_\beta). \tag{16}$$

---

[10]Or, for that matter, due to the obvious i.i.d nature of the algorithm, for any fixed $k$, the $k$-th output.

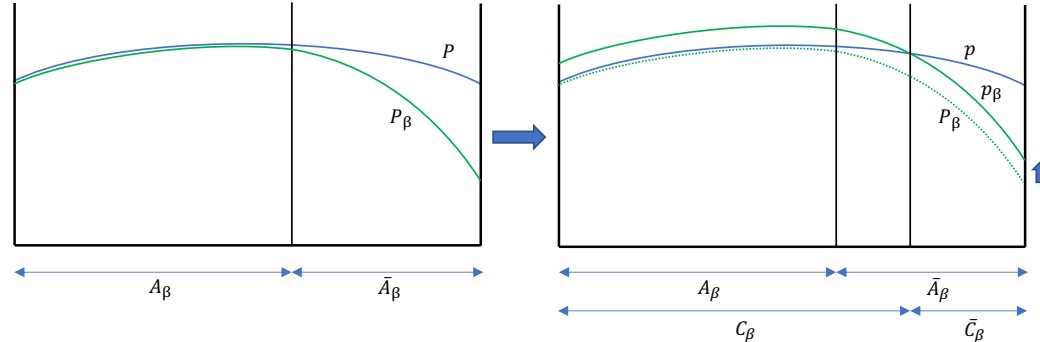

Figure 6: Visualization of Main Proof. The left panel shows the unnormalized distributions $P$ and $P_\beta$, the right panel their normalized versions $p$ and $p_\beta$. On the right panel, the area under the $p$ (resp. the $p_\beta$) curve represents the total $p$-mass (resp $p_\beta$-mass) of $X$, namely 1. To simplify visual comparison, the figure assumes that $Z = 1$, in other words that $P = p$; then $Z_\beta \leq 1$ and $p_\beta$ is $P_\beta$ moved up by the constant factor $\frac{1}{Z_\beta}$. The TVD between $p$ and $p_\beta$ is equal to the area between the two curves above $C_\beta$, but also to the area between the two curves above $\bar{C}_\beta$. This last area is included inside the area below the $p$ curve above $\bar{A}_\beta$, which is the visual counterpart of equation (16).

**Equation (4):** Clearly, for any (normalized) distribution $p$ over a discrete space $X$, for any $\epsilon > 0$, there exists a finite subset $X' \subseteq X$ s.t. $p(X') > 1 - \epsilon$. If we take $\beta \doteq \max_{x \in X'} \frac{P(x)}{q(x)}$, then $\beta$ is finite, $X' \subseteq A_\beta$, and therefore $p(A_\beta) \geq 1 - \epsilon$, which proves the result.

**A generalization to arbitrary $q$-supports** This last result exploits the assumption — that we made during the whole section 2 — that the support of $p$, $\mathrm{Supp}(p)$, is contained in the support of $q$, $\mathrm{Supp}(q)$, in other terms, $p(x) > 0 \Rightarrow q(x) > 0$. If that were not the case then $\beta \doteq \max_{x \in X'} \frac{P(x)}{q(x)}$ could be infinite and $A_\beta$ would not be defined. However, it is interesting that we can actually generalize the result to the case where $\mathrm{Supp}(p)$ may *not* be contained in $\mathrm{Supp}(q)$. Then, for any $\epsilon > 0$, there exists a finite subset $Y' \subseteq \mathrm{Supp}(q)$ s.t. $p(Y') > p(\mathrm{Supp}(q)) - \epsilon$. If we now take $\beta \doteq \max_{x \in Y'} \frac{P(x)}{q(x)}$, then $\beta$ is finite, $Y' \subseteq A_\beta$, and therefore $p(A_\beta) \geq p(\mathrm{Supp}(q)) - \epsilon$. For any $\beta$, because all the $x$'s in $A_\beta$ are obviously in $\mathrm{Supp}(q)$, we have $p(A_\beta) \leq p(\mathrm{Supp}(q))$, and therefore

$$p(\mathrm{Supp}(q)) \geq p(A_\beta) \geq p(\mathrm{Supp}(q)) - \epsilon. \tag{17}$$

When the support of $p$ is included in $\mathrm{Supp}(q)$, we have $p(\mathrm{Supp}(q)) = 1$, and therefore we get the previous result back, but now we see that, in the general case:

$$\lim_{\beta \to \infty} (1 - p(A_\beta)) = 1 - p(\mathrm{Supp}(q)). \tag{18}$$

## B TVD AND KL ESTIMATES

Here we provide derivations for the estimates of equations (9) and (10):

$$\mathrm{TVD}(p, p_\beta) = 1/2 \sum_{x \in X} |p(x) - p_\beta(x)| = 1/2\, \mathbb{E}_{x \sim q} \left| \frac{P(x)}{Zq(x)} - \frac{P_\beta(x)}{Z_\beta q(x)} \right| \tag{19}$$

$$\simeq 1/2\, N^{-1} \sum_{i \in [1,N]} \left| \frac{P(x_i)}{Zq(x_i)} - \frac{P_\beta(x_i)}{Z_\beta q(x_i)} \right|, \tag{20}$$

$$D_{\mathrm{KL}}(p, p_\beta) = \sum_{x \in X} p(x) \log \frac{p(x)}{p_\beta(x)} = \log \frac{Z_\beta}{Z} + \mathbb{E}_{x \sim q} \frac{P(x)}{Zq(x)} \log \frac{P(x)}{P_\beta(x)}, \tag{21}$$

$$\simeq \log \frac{Z_\beta}{Z} + N^{-1} \sum_{i \in [1,N]} \frac{P(x_i)}{Zq(x_i)} \log \frac{P(x_i)}{P_\beta(x_i)}. \tag{22}$$

Furthermore, we present the derivation and estimate for $D_{\mathrm{KL}}(p_\beta, a)$:

$$D_{\mathrm{KL}}(p_\beta, a) = \mathbb{E}_{x \sim p_\beta} \log \frac{p_\beta(x)}{a(x)} = \mathbb{E}_{x \sim p_\beta} \log \frac{P_\beta(x)}{Z_\beta \, a(x)} \tag{23}$$

$$= -\log Z_\beta + \mathbb{E}_{x \sim p_\beta} \log \frac{P_\beta(x)}{a(x)} \tag{24}$$

$$= -\log Z_\beta + \mathbb{E}_{x \sim q} \frac{P_\beta(x)}{Z_\beta \, q(x)} \log \frac{P_\beta(x)}{a(x)} \tag{25}$$

$$= -\log Z_\beta + Z_\beta^{-1} \, \mathbb{E}_{x \sim q} \frac{P_\beta(x)}{q(x)} \log \frac{P_\beta(x)}{a(x)} \tag{26}$$

$$\simeq -\log Z_\beta + Z_\beta^{-1} \, N^{-1} \sum_{i \in [1,N]} \frac{P_\beta(x_i)}{q(x_i)} \log \frac{P_\beta(x_i)}{a(x_i)}. \tag{27}$$

## C  PROJECTION OF THE PROPOSAL FOR POINTWISE CONSTRAINTS

Lets say we have a proposal distribution $q$ that we want to use for approximating an EBM with a pointwise constraint $P(x) = a(x)b(x)$ where $b(x) \in \{0, 1\}$. One simple approach would be projecting $q$ directly into the manifold of the sequences matching the constraint, by simply rejecting all sequences $x$ where $b(x) = 0$. In other words, we are sampling from a new probability distribution $q_{proj}(x) \propto q(x)b(x)$. It turns out that we can compute the probability assigned to each sequence $x$ by $q_{proj}$, as follows:

$$\begin{aligned} q_{proj}(x) &= 1/Z_{q_{proj}} \, q(x) && \text{for } b(x) = 1 \\ &= 0 && \text{for } b(x) = 0 \end{aligned}$$

Note that $Z_{q_{proj}} \doteq \sum_x q(x)b(x)$ can be easily estimated by using a sample from $q$.

Therefore, the $\mathrm{TVD}(p, q_{proj})$ and $D_{\mathrm{KL}}(p, q_{proj})$ can be estimated following the same procedure as Eqs. 19 and 21, respectively.

## D  INCREMENTAL PRUNING WITH MINIMAL EFFICIENCY TARGETS

The following algorithm describes a version of QRS that incrementally builds a batch of samples $S$ with a desired minimum acceptance rate $ar_{min}$. The algorithm works by obtaining samples $x$ from $q$ and provisionally storing them into $S$ as long as the rejection coefficient $\alpha_x = P(x)/q(x)u_x$ does not exceed the current value of $\beta$, where $u_x \sim \mathcal{U}(0, 1)$. Given that higher values of $\beta$ imply a lower upper bound on the TVD between the samples and the target distribution, $\beta$ is continually adapted to be as high as possible. Yet, the higher $\beta$ is, the lower the acceptance rate becomes. Therefore, to be practical, $\beta$ is capped at the highest possible value such that the acceptance rate of the samples seen so far would not fall below a minimum threshold $ar_{min}$. The corresponding maximum $\beta$ value for an acceptance rate threshold $ar_{min}$ is computed as the value $\alpha$ for which the percentage of all previously obtained samples satisfying $\alpha_x > \alpha$ is $ar_{min}$ (i.e., the percentile $ar_{min}$ of all previously computed $A = \{\alpha_x\}$ values). Should $\beta$ be increased at any point, previously stored samples in $S$ are pruned to remove those that failed to meet the acceptance criterion $\alpha_x > \beta$. The algorithm stops when at least $n$ samples have been collected.

---

**Algorithm 2** QRS with incremental pruning

---

**Require:** EBM $P$, proposal $q$, desired number of samples $n$, minimum acceptance rate $ar_{min}$.
**Ensure:** A set of $n$ samples $S$ and an estimation of $\beta$ s.t. the acceptance rate is at least $ar_{min}$.
 1: **procedure** QRS-INCREMENTAL
 2:     $S \leftarrow [\,], \beta \leftarrow 0, A \leftarrow [\,], \beta_{max} \leftarrow 0$
 3:     **while** $|S| < n$ **do**
 4:         $x \sim q(.)$
 5:         $\beta_x \leftarrow \frac{P(x)}{q(x)}$
 6:         $\beta_{max} \leftarrow \text{PERCENTILE}(A, ar_{min})$
 7:         **if** $\min(\beta_x, \beta_{max}) > \beta$ **then**
 8:             $\beta \leftarrow \min(\beta_x, \beta_{max})$
 9:             $S \leftarrow \text{PRUNE}(S, \beta)$
10:         $u_x \sim \text{Unif}(0, 1)$
11:         $\alpha_x \leftarrow \frac{\beta_x}{u_x}$
12:         **if** $\alpha_x > \beta$ **then**                                                    ▷ i.e. $u_x < \frac{P(x)}{\beta q(x)}$
13:             $S \leftarrow S + [(x, \alpha_x)]$
14:         $A \leftarrow A + [(\alpha_x)]$
15:     **return** $S, \beta$
16: **procedure** PRUNE$(S, \beta)$
17:     keep $(x, \alpha_x)$ in $S$ iff $\alpha_x > \beta$                                        ▷ i.e. $u_x < \frac{P(x)}{\beta q(x)}$
18: **procedure** PERCENTILE$(A, r)$
19:     return max $\alpha_x$ in $A$ s.t. $|\alpha_y \in A : \alpha_y \leq \alpha_x| < r|A|$.

---

# E  INDEPENDENT METROPOLIS HASTINGS (IMH): ALTERNATIVE TO QRS ?

274     7 The Metropolis–Hastings Algorithm

We therefore have the following convergence result for Metropolis–Hastings Markov chains.

**Theorem 7.4.** *Suppose that the Metropolis–Hastings Markov chain $(X^{(t)})$ is $f$-irreducible.*

*(i) If $h \in L^1(f)$, then*

$$\lim_{T \to \infty} \frac{1}{T} \sum_{t=1}^{T} h(X^{(t)}) = \int h(x) f(x) dx \qquad a.e.\ f.$$

*(ii) If, in addition, $(X^{(t)})$ is aperiodic, then*

$$\lim_{n \to \infty} \left\| \int K^n(x, \cdot) \mu(dx) - f \right\|_{TV} = 0$$

*for every initial distribution $\mu$, where $K^n(x, \cdot)$ denotes the kernel for $n$ transitions, as in (6.5).*

Figure 7: Metropolis-Hastings: Theorem 7.4. copied from (Robert & Casella, 2004). Here $f$ is the target distribution. Note the difference between (i) and (ii). Property (i) is concerned with the $f$-expectation of R.V. $h$, and considers the average over the $T$ first elements of a single chain, which converges to the expectation for increasing $T$. Property (ii) is concerned with the TVD distance between the target distribution $f$ and the distribution obtained by repeatedly running an $n$-step chain and outputting the $n$-th element. This distance converges to 0 for increasing $n$.

## E.1  THE INDEPENDENT METROPOLIS HASTINGS (IMH) ALGORITHM

Markov chain Monte-Carlo (MCMC) (Robert & Casella, 2004) is the most developed general class of techniques for sampling from EBMs, exploiting random walk and Markov Chain theory (rejection

sampling (RS) and QRS do not use random walks, and do not qualify as MCMC). In the situation that we are focussing on here, namely, the availability of a global proposal $q$ that already approximates $p$ to some extent, a suitable MCMC technique is *Independent Metropolis-Hasting* (IMH) (Robert & Casella, 2004, §7.4), see Algorithm 3.

---

**Algorithm 3** IMH

---

**Require:** $P, q$
1: $x \sim q$
2: **while** True **do**
3:     $x' \sim q$
4:     $r_{x,x'} \leftarrow \min(1, \frac{P(x')/q(x')}{P(x)/q(x)})$                                    ▷ Prob. of moving to $x'$
5:     $u \sim U_{[0,1]}$
6:     **if** $u \leq r_{x,x'}$ **then**
7:         output $x'$
8:         $x \leftarrow x'$
9:     **else**
10:         output $x$

---

IMH is a special case of the Metropolis-Hastings (MH) algorithm, where the proposal $q(x')$ does not depend on the current $x$.[11]

**IMH and QRS vs. RS: no need for global $\beta$**    IMH and QRS share a common advantage over RS: they do not need a global $\beta$ with $P(x)/q(x) \leq \beta, \forall x \in X$. It is often the case that such a bound does not exist, is not known, or is intractably large. In such cases, RS cannot be used, but both IMH and QRS can.

**IMH vs. QRS: convergence to $p$ in the limit**    IMH inherits from MH a fundamental "convergence to $p$ in the limit" property:

$$\lim_{n \to \infty} \text{TVD}(p, \pi^{(n)}) = 0 \tag{28}$$

where $\pi^{(n)}$ denotes the sampler associated with what we will call the "$n$-steps variant of Algorithm 3", namely the algorithm that performs the loop on Line 1 exactly $n$ times, does not output anything, but returns the last $x$. This property corresponds, in our own notation, to Theorem 7.4.(ii) of (Robert & Casella, 2004), copied in Fig. 7.

QRS has its own form of convergence in the limit, Equation 4.

**IMH vs. QRS: convergence diagnostics**    Equation 28 tells us that performing $n$ steps of MH (and in particular IMH) produces a distribution $\pi^{(n)}$ that gets closer and closer to $p$ with increasing $n$, but it does not provide any explicit estimate of $\text{TVD}(p, \pi^{(n)})$, nor any other divergence metric from the target distribution.

In general, with MCMC techniques, such explicit convergence diagnostics are very difficult to obtain (Cowles & Carlin, 1996; Roy, 2020). By contrast, as we saw (Equations 3, 7, 9, 10), the QRS algorithm does provide such *explicit estimates*. This is important because it allows us to calibrate the level of effort (acceptance rate) we have to invest in order to obtain a certain quality (TVD or KL to $p$).[12]

**IMH vs. QRS: ability to score**    Equation (1) provides an explicit value for $p_\beta(x)$. This is an important property, which, in particular, is exploited in the computation of estimates of $\text{TVD}(p, p_\beta)$ and $D_{\text{KL}}(p, p_\beta)$ in Equations (9) and (10). This ability to score is conspicuously absent in IMH: there is no obvious way to compute $\pi^{(n)}(x)$, not even up to a constant factor.

---

[11]In the general MH algorithm, line 4 is replaced by: $r_{x,x'} \leftarrow \min(1, \frac{P(x')/q(x'|x)}{P(x)/q(x|x')})$.

[12]However, IMH does have an advantage over MH here. In the special case that a global $\beta$ s.t. $\frac{p(x)}{q(x)} \leq \beta, \forall x \in X$ exists, it can be shown that $\text{TVD}(p, \pi^{(n)}) \leq 2(1 - \beta^{-1})^n$ (Robert & Casella, 2004, p. 277). However, this bound can be very conservative and unusable in practice, even in those cases where it is explicitly known.

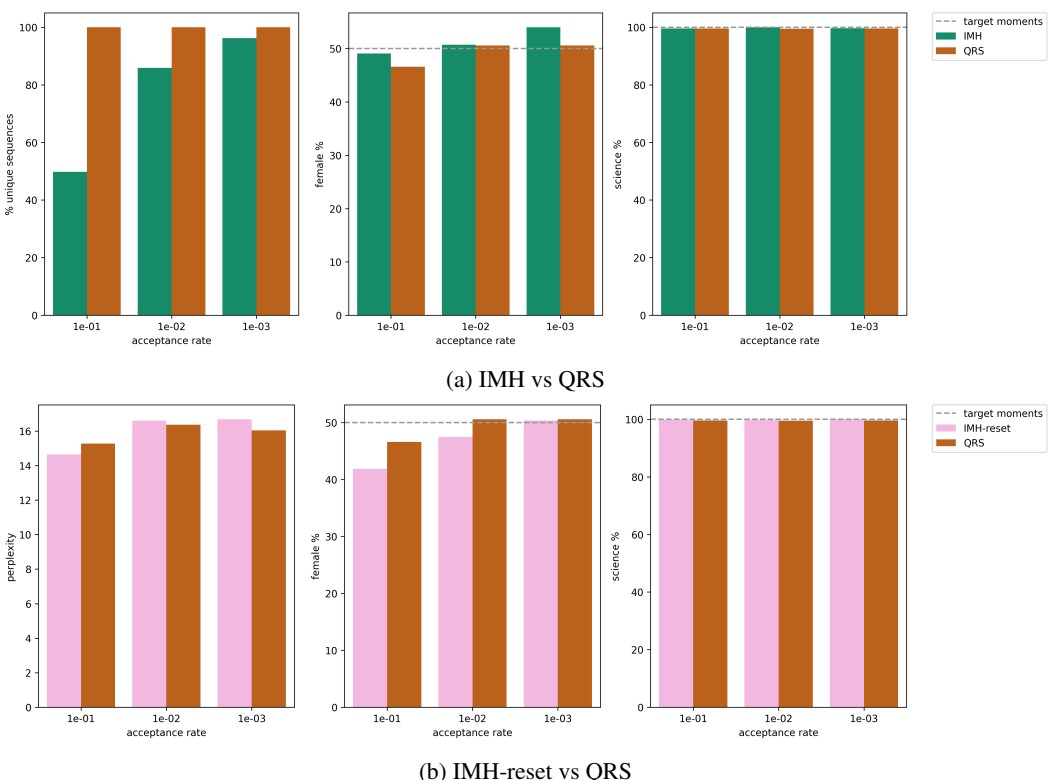

(a) IMH vs QRS

(b) IMH-reset vs QRS

Figure 8: We compare the QRS and IMH samplers in practice by taking 1,000 samples at 3 orders of acceptance rate for the constraint 50% female and 100% science. We experiment with a version of IMH in which we use a fixed burn-in of 1000 and set thinning as to obtain the desired acceptance rate, as well as a version in which we reset the chain after 10, 100, or 1,000 samples. For the latter version we can also estimate perplexity as the samples are i.i.d. We do not show the percentage of unique samples for the latter version, but note this is 100% for both QRS and IMH-reset.

**IMH vs. QRS: i.i.d properties**  The outputs produced by the QRS Algorithm 1 are immediately i.i.d: the acceptance of an $x$ on Line 6 does not depend on the acceptance of the previous $x$. By contrast, with any MH algorithm, and in particular IMH, if the goal is to produce actual samples, we face a choice. First, we could follow the spirit of (Theorem 7.4.ii) above, restarting the chain each time we need to produce a sample. Then we would produce i.i.d samples, but at the high cost of wasting $n-1$ draws from the proposal before producing just one sample. Or we can — similar to the spirit of (Theorem 7.4.i) for expectations — just produce one long chain from which we extract the actual samples, which is the approach we are taking in Algorithm 3. The outputs of this algorithm are not i.i.d: it is possible for the new proposal $x'$ to be rejected (Line 6) and then the current $x$ is repeated in the sequence of outputs, possibly many times if the current $x$ is "good" (i.e. has a high importance ratio $\frac{P(x)}{q(x)}$). To mitigate that problem, it is often advocated to use a "thinning" heuristics on the outputs of the algorithm: only retain one output out of $m$, in order to reduce the autocorrelation, in the hope of obtaining a good balance between the quality of the samples and the efficiency of the sampler, but again, without explicit quality estimates.

### E.2  IMH VS. QRS EXPERIMENT

In Figure 8a we compare the QRS and the IMH sampler on the 50% female and 100% science EBM by taking 1,000 samples from each sampler at different levels of acceptance rates. We use the procedure described in Appendix D to obtain QRS samples at the specified acceptance rate. For the IMH sampler we obtain the desired "acceptance rate" by having a fixed burn-in period of 1,000 samples and afterwards only keep every $10_{th}$, $100_{th}$ or $1000_{th}$ sample. The definition of "acceptance rate" we use for both samplers is the ratio of samples obtained from the sampler (1,000)

and the number of samples used from the proposal distribution. We use the DPG model as proposal distribution.

We find that while IMH is able to achieve constraint satisfaction similar to the QRS sampler at similar acceptance rates, it does so by repeating some samples multiple times. This can be seen in the fraction of unique samples, which can get as low as 50% out of the 1,000 samples obtained at lower amounts of thinning. While this does not hinder the IMH sampler from achieving asymptotic consistency in estimating expectation values, we believe this to be an undesirable property of a sampler whose outputs are to be used in actual applications. QRS does not suffer from the same problem, as the QRS algoritm does not make use of a Markov chain.

In Figure 8b we also experiment with a version of IMH that does obtain i.i.d. samples by resetting the Markov chain after every sample obtained. We coin this version IMH-reset. We achieve desired acceptance rates by resetting the chain after every $10_{th}$, $100_{th}$ or $1000_{th}$ sample. We obtain 100% unique sequences for this version for a sample of 1,000 samples, like the QRS sampler. For this version we can additionally estimate perplexity under GPT-2, as we have i.i.d. samples from both samplers. We find that at the target feature moments (for both samplers at the acceptance rate of $10^{-3}$, QRS samples achieve slightly lower perplexity than IMH-reset.

## F    ADDITIONAL RELATED WORK: CONTINUOUS EBMS AND EBMS FOR NLP

While this paper is concerned about *generating* samples from *discrete* EBMS, more research so far has been more concerned with the *training* of *continuous* EBMs, in particular for applications in vision. Continuous EBMs have the advantage over discrete ones in that it is possible to differentiate the EBM $P(x)$ relative to $x$, and not only the approximating model $\pi_\theta$ relative to $\theta$. Training such EBMs (see the survey in (Song & Kingma, 2021)) can then often be addressed through techniques such as *contrastive divergence* (Hinton, 2002; Du et al., 2021), *score matching* (Hyvärinen, 2005; Song & Ermon, 2020), or *noise contrastive estimation* (Gutmann & Hyvärinen, 2010). These approaches sometimes require an internal sampling procedure, and then one technique of choice is *Langevin MCMC* (Parisi, 1981), in which the local Markov chain moves are done based on $\nabla_x P(x)$, a technique which is also employed in case actual samples need to be generated. While such techniques are not available for discrete EBMs, some recent efforts are trying to bridge the gap (Grathwohl et al., 2021).

Sampling techniques are popular in various natural language processing (NLP) applications. For example, Miao et al. (2020) construct a rejection-sampling inspired sampler that aims to counteract over- and underestimation of probability regions due to overfitting when fine-tuning large pre-trained language models on small datasets. Deng et al. (2020) train globally normalized language models to combat negative effects of local normalization, and use a form of SIR to sample from the resulting EBM using an autoregressive proposal language model. Goyal et al. (2021) develop a Metropolis-Hastings algorithm to sample from masked language models. For controlled text generation Miao et al. (2019) propose a random-walk Metropolis-Hastings (MH) algorithm for sampling from an EBM that encodes sequence-level preferences on natural text. Their proposal distribution consists of local string editing operations on randomly selected words or positions. Zhang et al. (2020) improve on this approach by making use of a tree-search algorithm to more efficiently explore the space of proposals, by allowing several edits in a single step of the MH algorithm. In contrast with these approaches we make use of a strong global proposal distribution and the QRS sampler. This eliminates any autocorrelations in the samples and ensures good sample diversity inherited from the underlying EBM. Furthermore, the QRS sampler allows us to assess the quality of our samples through divergence metrics from the target EBM, something that is famously difficult to do with MCMC samplers.

## G    ADDITIONAL CONTROLLED TEXT GENERATION RESULTS

We perform constrained text generation on a distributional, two pointwise and two mixed distributional pointwise constraints. In particular, we constrain the GPT-2 biographies model to contain 1) 50% female biographies about scientists, 2) 50% female biographies about sports or 3) 50% female biographies without additional constraint. Also, we constrain GPT-2 small to generate exclusively sequences containing the term "amazing" or to generate exclusively sequences containing the term "Wikileaks". For each of these tasks, we obtained a fine-tuned model using DPG, which serves both

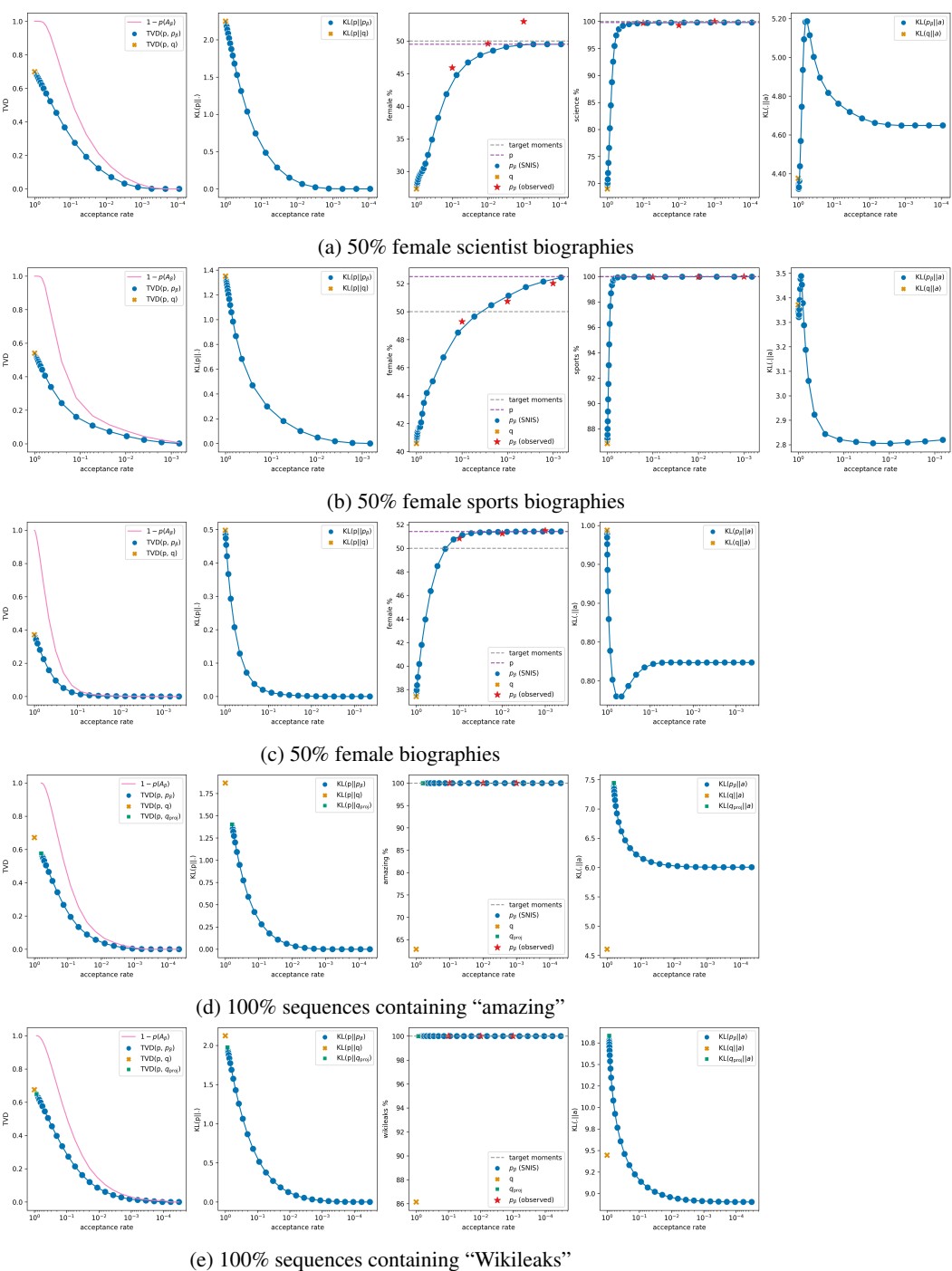

(a) 50% female scientist biographies

(b) 50% female sports biographies

(c) 50% female biographies

(d) 100% sequences containing "amazing"

(e) 100% sequences containing "Wikileaks"

Figure 9: We show importance sampling estimates of $\text{TVD}(p, .)$, an upper-bound on $\text{TVD}(p, p_\beta)$, $D_{\text{KL}}(p||.)$, $D_{\text{KL}}(.||a)$ and feature moments as a function of acceptance rate. We show three distributional constraints on GPT-2 Biographies and two pointiwse constraints on GPT-2 small. As proposal distribution we make use of a DPG model trained for each constraint separately. We show separate lines for the target moments and the moments realized by the EBMs, revealing slight inaccuracies in the EBM moments for some constraints. We also show observed moments for 50k QRS samples obtained at acceptance rates $10^{-1}$, $10^{-2}$, and $10^{-3}$.

| Experiment | $\beta_{\min}$ | $\beta_{\max}$ |
|---|---|---|
| 50% female and 100% scientists | $1.0 \cdot 10^{-12}$ | $9.3 \cdot 10^{6}$ |
| 50% female and 100% sports | $1.0 \cdot 10^{-12}$ | $2.9 \cdot 10^{7}$ |
| 50% female | $4.0 \cdot 10^{-7}$ | $4.0 \cdot 10^{3}$ |
| 100% "amazing" | $1.0 \cdot 10^{-12}$ | $5.3 \cdot 10^{1}$ |
| 100% "Wikileaks" | $1.0 \cdot 10^{-12}$ | $6.0 \cdot 10^{0}$ |

Table 1: We report the range of $\beta$ values used to obtain the range of acceptance rates in Figure 9. We note that the procedure described in Appendix D can be used to find $\beta$ values that target some minimum acceptance rate.

as a baseline and as a proposal $q$ that we can sample from. In the case of pointwise constraints, we also consider a *naive filter* sampler $q_{proj}$ in which the proposal distribution is directly projected onto the constraint manifold by filtering out all samples that do not match the constraint. This sampler also assigns well-defined probabilities to the sequences that it samples (see Appendix C), and so we can compute estimates of the TVD and $D_{\mathrm{KL}}$ for it.

For each task, we again obtain 1M samples from the corresponding proposal, which we use to evaluate the proposal $q$, the projected proposal $q_{\mathrm{proj}}$ (only for the pointwise constraint), and QRS sampling ($p_\beta$) for a range of $\beta$ values reported in Table 1. For all of these, we compute estimates of the same metrics as in Section 3.2.2 (i.e., $\mathrm{TVD}(p, p_\beta)$, $D_{\mathrm{KL}}(p, p_\beta)$, AR, backward KL-divergence from the base language model $D_{\mathrm{KL}}(\cdot||a)$, and the moments of the features that we wish to control). Furthermore, we do a downstream evaluation of 50k samples obtained through QRS choosing $\beta$ so that the acceptance rate falls exactly at $10^{-1}$, $10^{-2}$ and $10^{-3}$. (The process is described in Appendix D). In particular, we look at the feature moments of the obtained samples.

Our results are shown in Fig. 9. As expected, we find that the upper bound of the TVD of $p_\beta$ with $p$ and the KL-divergence from $p_\beta$ to $p$ steadily converge towards 0 as the acceptance rate decreases. For the distributional constraints and corresponding proposal distributions shown here, it seems that an acceptance rate of $10^{-3}$ is sufficient to match the target EBM nearly perfectly. Feature moments therefore shows the same pattern, converging towards the target moments with lower acceptance rates, although we find that in some cases the QRS sampler matches the target EBM so closely that small inaccuracies in the lambdas obtained from the EBM estimation procedure as described in Khalifa et al. (2021) become apparent. As for the divergence to the original language model $D_{\mathrm{KL}}(p_\beta||a)$, there is no obvious trajectory that it should follow other than it should converge to the lowest possible value when all constraints are satisfied. Indeed, our results show that this metric follows a non-monotonic path at different ARs. Validating our estimates, we note that the moments computed downstream on QRS match tightly the IS predictions, giving us confidence in the accuracy of those estimates. Finally, in the case of our pointwise "amazing" and "Wikileaks" constraints, we find that the naive filter strategy ($q_{\mathrm{proj}}$) corresponds to running the QRS sampler at a high acceptance rate.

## H  EXTRA EXPERIMENTS AND COMPARISONS WITH LOCALIZED MCMC

We focus on discrete Random Walk Metropolis-Hastings (RWMH) for text generation, conceptually close to Miao et al. (2019) with some ingredients from Goyal et al. (2021). We did consider GwG (Gibbs-with-Gradients Grathwohl et al. (2021)) and other related gradient-based proposals, but felt that attempting to apply such techniques for text generation tasks such as those considered in our main experiments would be very ambitious and worth a paper on its own, see Appendix I.

In our proposed experiments, we construct a local proposal distribution composed of single-token insert, delete and replace operations (chosen randomly from a uniform distribution like in Miao et al. (2019)). We inform the insert (resp. replace) operation by inserting (resp. replacing a word with) a [MASK] token and sampling a new token from BERT. This differs from Miao et al. (2019), who instead do a form of Gibbs sampling for insert and replace operations, which requires V (vocabulary size) assessments of the EBM per such operation (an EBM with GPT-2 as a component). This would make this version prohibitively more expensive than our QRS and IMH operations, for which reason we replace this with a single pass of BERT instead. We accept or reject such edits using the Metropolis-Hastings acceptance ratio and seed the chain using a high-quality global proposal

| Method | AR | %Female (exp. 50%) | %Science (exp. 100%) | PPL↓ | Self-BLEU-5↓ | %Uniq↑ | TVD↓ | KL↓ |
|---|---|---|---|---|---|---|---|---|
| GDC | 1 | $28.1 \pm 1.8$ | $69.2 \pm 1.2$ | $34.9 \pm 1.1$ | $89.9 \pm 0.2$ | $100 \pm 0.0$ | tbd | tbd |
| RWMH-10 | $10^{-1}$ | 37.5 | 32.2 | – | 99.97 | 23.4 | Unk. | Unk. |
| RWMH-reset-10 | $10^{-1}$ | 26.3 | 68.4 | 36.1 | 89.6 | 100 | Unk. | Unk. |
| IMH-10 | $10^{-1}$ | 55.0 | 100 | – | 94.7 | 58.0 | Unk. | Unk. |
| IMH-reset-10 | $10^{-1}$ | 41.4 | 99.4 | 29.6 | 91.4 | 100 | Unk. | Unk. |
| QRS | $10^{-1}$ | $44.1 \pm 1.1$ | $99.8 \pm 0.2$ | $30.2 \pm 0.8$ | $91.0 \pm 0.2$ | $100 \pm 0.0$ | 0.29 | 0.56 |
| RWMH-1000 | $10^{-3}$ | 0 | 100 | – | 100 | 0.1 | Unk. | Unk. |
| RWMH-reset-1000 | $10^{-3}$ | 26.6 | 68.2 | 33.1 | 90.0 | 100 | Unk. | Unk. |
| IMH-1000 | $10^{-3}$ | 51.5 | 99.9 | – | 91.2 | 94.9 | Unk. | Unk. |
| IMH-reset-1000 | $10^{-3}$ | 48.8 | 99.8 | 35.6 | 90.8 | 100 | Unk. | Unk. |
| QRS | $10^{-3}$ | $49.3 \pm 1.5$ | $99.8 \pm 0.1$ | $34.6 \pm 1.1$ | $90.8 \pm 0.1$ | $100 \pm 0.0$ | 0.02 | 0.02 |

Table 2: Additional results comparing MC sampling methods on obtaining samples from the "female-science" EBM described in Section 3.2.2. We do not compute perplexity for IMH and RWMH without reset as it does not yield i.i.d. samples. As noted, TVD and KL for MCMC methods are unknown (i.e. we have no way of estimating them). Where available we show mean $\pm$ one standard deviation over 10 runs.

| Method | AR | %Amazing (exp. 100%) | PPL↓ | Self-BLEU-5↓ | %Uniq↑ | TVD↓ | KL↓ |
|---|---|---|---|---|---|---|---|
| GDC | 1 | $63.0 \pm 1.5$ | $62.4 \pm 1.2$ | $85.7 \pm 0.3$ | $100 \pm 0.0$ | tbd | tbd |
| RWMH-10 | $10^{-1}$ | 100 | - | 99.96 | 47.9 | Unk. | Unk. |
| RWMH-reset-10 | $10^{-1}$ | 61.7 | 65.1 | 85.3 | 100 | Unk. | Unk. |
| IMH-10 | $10^{-1}$ | 100 | - | 91.6 | 62.9 | Unk. | Unk. |
| IMH-reset-10 | $10^{-1}$ | 100 | 60.4 | 87.2 | 100 | Unk. | Unk. |
| QRS | $10^{-1}$ | $100 \pm 0.0$ | $62.8 \pm 1.1$ | $86.9 \pm 0.3$ | $100 \pm 0.0$ | 0.17 | 0.27 |
| RWMH-1000 | $10^{-3}$ | 71.2 | - | 97.3 | 59.2 | Unk. | Unk. |
| RWMH-1000 + 0.01 * $\delta$("amazing") | $10^{-3}$ | 100 | – | 100.0 | 0.1 | Unk. | Unk. |
| RWMH-1000 + 0.1 * $\delta$("amazing") | $10^{-3}$ | tbd | tbd | tbd | tbd | Unk. | Unk. |
| RWMH-reset-1000 | $10^{-3}$ | 62.9 | 63.7 | 85.5 | 100 | Unk. | Unk. |
| RWMH-reset-1000 + 0.01 * $\delta$("amazing") | $10^{-3}$ | 64.1 | 61.6 | 85.8 | 100 | Unk. | Unk. |
| RWMH-reset-1000 + 0.1 * $\delta$("amazing") | $10^{-3}$ | 63.7 | 59.9 | 85.5 | 100 | Unk. | Unk. |
| IMH-1000 | $10^{-3}$ | 100 | - | 87.3 | 99.6 | Unk. | Unk. |
| IMH-reset-1000 | $10^{-3}$ | 100 | 64.0 | 87.1 | 100 | Unk. | Unk. |
| QRS | $10^{-3}$ | $100 \pm 0.0$ | $64.1 \pm 1.3$ | $86.6 \pm 0.2$ | $100 \pm 0.0$ | 0.01 | 0.01 |

Table 3: Additional results comparing MC sampling methods on the task of obtaining samples from an EBM with a point-wise constraint to include the word "amazing" in the sequence. We do not compute perplexity for IMH and RWMH without reset as it does not yield i.i.d. samples. As noted, TVD and KL for MCMC methods are unknown (i.e. we have no way of estimating them). Where available we show mean $\pm$ one standard deviation over 10 runs.

distribution (DPG, as used in QRS experiments). Per step of the MCMC chain this algorithm is roughly similar in cost to our IMH algorithm. A single step of RWMH requires a pass through BERT (for insert or replace) and a pass through GPT-2 (for scoring under the EBM), while a single step of IMH requires sampling a sequence from DPG (a highly parallelizable operation, in practice we take many samples at once and pre-store them on disk) and pass through GPT-2 as well (for scoring under the EBM). Therefore, we use the same definition of acceptance rate as in the IMH experiments in Appendix E. Like for IMH, we also provide two versions of RWMH: one in which we do thinning to reduce autocorrelations (RWMH) and one in which we reset the chain (RWMH-reset) after a number of steps (and set the seed again from the DPG proposal). The value of the thinning parameter and the number of steps after which the chain is reset determines the AR of the algorithm. In order to inform the local proposal distribution of the EBM, we also investigate mixing BERT with a Dirac $\delta$ on a particular word (e.g. "amazing" or "Wikileaks") for a pointwise constraint. We compare each sampler (QRS, IMH, IMH-reset, RWMH, RWMH-reset) in terms of moment matching results, perplexity, Self-BLEU (Zhu et al., 2018), and finally TVD and KL where possible (that is to say, for QRS only, as we stress below).

We run on both a pointwise constraint (generating from GPT-2 using the word "amazing") and a mixed distributional constraint (debiasing scientist biographies, see Section 3.2.2) to compare the samplers. We run all samplers (QRS, IMH, IMH-reset, RWMH, RWMH-reset) both at $10^{-1}$ and $10^{-3}$ acceptance rate. We use Algorithm 2 for QRS to find a $\beta$ value that approximately satisfies the target acceptance rate. For IMH and RWMH we have a fixed burn-in period of 1,000 steps and define the acceptance rate as the amount of thinning that is done: e.g. keep only every $10_{th}$ sample for an acceptance of $10^{-1}$. For IMH-reset and RWMH-reset we define the acceptance rate by the rate at

which we reset the chain (and re-seed from the global proposal), e.g. we run a chain of length 1,000 and only keep the last sample for an acceptance rate of $10^{-3}$. We also report performance on the global proposal (DPG). We show results on collecting 1,000 samples of each in Tables 2 and 3.

We find that IMH, IMH-reset and QRS perform considerably better than RWMH and RWMH-reset at all acceptance rate levels. RWMH and RWMH-reset seem to have trouble achieving good constraint satisfaction consistently. We currently report numbers on a single run of the sampler, which might suffer from a poor seed sampled from DPG. In the final version we shall include mean and variance results over multiple runs.

QRS, IMH and IMH-reset meet the imposed constraint of only generating sequences containing the term "amazing" and, at lower acceptance rate, also the constraints of generating debiased scientist biographies roughly. At $10^{-1}$ acceptance rate IMH suffers from high autocorrelation among the samples due to repetitions (as seen from the percentage of unique sequences). Also at a $10^{-3}$ acceptance rate IMH and IMH-reset both seem to have slightly less diversity than QRS, but all in all do achieve similar self-BLEU as QRS and produce roughly 1,000 unique sequences. Perplexity is also similar for QRS and IMH-reset (we do not compute perplexity for IMH as the computation requires i.i.d. samples), slightly in favor of QRS in the mixed distributional constraint. We found similar results before in Appendix E.2.[13] In conclusion, QRS and IMH-reset have an advantage over IMH in that they provide i.i.d. samples and have higher diversity at high acceptance rates. Both versions of IMH and QRS perform on par in terms of perplexity and self-BLEU. QRS, however, has a large advantage in that it allows to compute actual divergence from the target distribution (TVD and KL), something simply not available to MCMC methods.[14]

### H.1 Informing the Local Proposal Distribution

We attempt to inform the local proposal distribution without introducing excessive additional computation by proposing a simple mixture model for the insert and replace operations. With a small probability, that we tune as a hyperparameter, the proposal distribution proposes to insert or replace a token with the token "amazing" (additionally to the probability of that occurring under BERT alone). We show some preliminary results of this on RWMH and RWMH-reset in Table 3.

Our preliminary results do generally increase the constraint satisfaction of the resulting sampler. However, in the case of RWMH it leads to a catastrophic failure mode: where in the burn-in period an "amazing" was inserted somewhere in the sequence, and the chain got stuck at that local optimum for all 1,000 samples we collected (including thinning steps). The results on RWMH-reset look more promising, increasing constraint satisfaction only slightly, but steadily improving perplexity. We will perform more experiments on this for the camera-ready version and include some variance estimates over different runs of the algorithm.

## I Observations on GwG and other gradient-based techniques for discrete spaces

Gibbs with Gradients (GwG) (Grathwohl et al., 2021) is a recent promising technique for importing the effective gradient techniques of continuous EBMs (as in vision) to discrete EBMs. Grathwohl et al. (2021) start by observing that basic techniques for discrete distributions such as RWMH (Random Walk Metropolis Hastings) (i) require meticulous customization of the local proposal to lead to reasonable results and (ii) are unable to exploit differentiability over sample-space densities, which are so crucial for efficient MCMC sampling from EBMs in continuous domains. The method they propose instead addresses those situations (which they argue are many) where the discrete EBM can be seen as a projection of a differentiable function over an underlying continuous space, which

---

[13]We note that our perplexity numbers slightly differ in magnitude compared to Appendix E.2 due to a slight change in tokenization. We will update this in the final version to be consistent throughout the appendices.

[14]Note that constraint satisfaction plus perplexity estimates *do not* provide a good estimate of distributional divergence (TVD or KL) from $p$. This is true for both pointwise and distributional constraints. For instance, with the female-science experiment, if a sampler $\omega$ generates only two fluent sentences each with probability 0.5, both about science, and one of the two with a female mention, then constraint satisfaction is perfect and perplexity (i.e. fluency) is very good, but divergence from $p$ can be terrible. Self-BLEU is a proxy to diversity that improves the evaluation somewhat, but still does not allow to estimate divergence.

| $\beta$ | 1e$-$12 | 3.1e3 | 4.4e4 | 6.4e5 | 9.3e6 |
|---|---|---|---|---|---|
| AR | $1.00 \pm 0.00$ | $1.4e{-}1 \pm 3.4e{-}4$ | $1.6e{-}2 \pm 6.8e{-}5$ | $1.3e{-}3 \pm 1.2e{-}5$ | $9.2e{-}5 \pm 1.8e{-}6$ |
| $\mathrm{TVD}(p, p_\beta)$ | $0.70 \pm 0.00$ | $0.38 \pm 0.01$ | $0.14 \pm 0.01$ | $0.02 \pm 0.01$ | $1.6e{-}6 \pm 5.6e{-}7$ |
| $D_{\mathrm{KL}}(p, p_\beta)$ | $2.35 \pm 0.09$ | $0.83 \pm 0.08$ | $0.21 \pm 0.05$ | $0.02 \pm 0.02$ | $3.3e{-}7 \pm 2.0e{-}6$ |

Table 4: Means and standard deviation of importance sampling estimates of acceptance rate, TVD with the target distribution and KL-divergence to the target distribution for various $\beta$ on debiasing scientist biographies (see Section 3.2.2). We perform 10 runs using 1,000,000 samples each to compute the means and standard deviations. Values of $\beta$ are chosen within the range used for our experiments as reported in Table 1.

leads them to perform sampling in the discrete space, but with a novel "locally informed proposal" (Zanella, 2017) that exploits gradients in the underlying continuous space in order to avoid explicitly computing the energy of each point in the "local neighborhood" of $x$, a very costly operation.

Grathwohl et al. (2021) do not experiment with textual data, and we believe that while applying their technique to some text generation EBMs similar to those of our current experiments would be possible (see just below), this would be an ambitious research project on its own, beyond the scope of the current paper.

A possible GwG approach for $P(x)$, where $x = x_1, ..., x_n$ is a text to be generated, and where $P(x)$ could be computed as a differentiable function of $e_1, ..., e_n$, where $e_i$ is the embedding of $x_i$, could work as follows, adapting to text generation the last line ("Deep EBM") of Table 1 in Grathwohl et al. (2021). We would first pick some position $m$, then compute $\nabla_{e_m} P(e_1, ..., e_m, ..., e_n)$. Then for each $x' = x_1, ..., x'_m, ..., x_n$ differing from $x$ only at position $m$, we would approximate $P(e_1, ..., e'_m, ..., e_n)$ using the previous gradient,[15] a much less costly operation than computing $P(e_1, ..., e'_m, ..., e_n)$ from scratch (this is the gist of GwG). In principle such a procedure could be done for any $P(x) = a(x)b(x)$, where $a$ is a neural model (hence in principle differentiable over the word embeddings, e.g. for $a$ an autoregressive model), and $b$ is differentiable over these embeddings. As mentioned above, this would be a whole project on its own, but note that the requirement that $b$ be differentiable could make the approach tricky to apply to some cases such as when $b$ computes an arbitrary binary predicate, is provided as a black box, etc.

Grathwohl et al. (2021) contrast their approach to other recent gradient-based techniques, such as "continuous relaxation", that first map discrete $x$'s to continuous representations, perform gradient-based MCMC there (e.g. with HMC, SGLD, etc.) and then map back to the discrete space (while GwG performs MCMC in the discrete space). They argue that MCMC sampling in the continuous relaxation does not directly exploit the underlying discrete structure of the space and therefore that sampling performance in the relaxed space may not be indicative of sampling performance in the discrete space. We are not aware of such techniques that we could directly apply to our text generation tasks, and did not attempt experiments in this space.

## J   IMPORTANCE SAMPLING AND VARIANCE

We perform many importance sampling estimates in this work to estimate quantities such as acceptance rate, and TVD and KL to the target distribution. We report variance estimates for the experiments on debiasing scientist biographies (Section 3.2.2) in Table 4. We collect 10 times the number of samples used to generate our plots (1 million) and report mean $\pm$ one standard deviation for $\beta$ values within the range used within our experiments (also see Table 1). We find our estimates to be accurate within reasonable variance.

---

[15] $P(e_1, ..., e'_m, ..., e_n) \simeq P(e_1, ..., e_m, ..., e_n) + \langle \nabla_{e_m} P(e_1, ..., e_m, ..., e_n), (e'_m - e_m) \rangle$.

