# OpenReview forum: "Sampling from Discrete Energy-Based Models with Quality/Efficiency Trade-offs"
_ICLR.cc/2022/Conference — ICLR 2022 Submitted_

### Official Review · Reviewer_d6gm · 2021-10-25

**Correctness:** 4
**Technical Novelty And Significance:** 3
**Empirical Novelty And Significance:** 3
**Recommendation:** 6
**Confidence:** 4

**Details Of Ethics Concerns:**

None.

**Main Review:**

Sampling from unnormalized distributions is notoriously difficult. Traditional rejection sampling cannot be applied in many settings (including the settings addressed in this work) and because of this, MCMC techniques are often favored. Unfortunately, diagnostics for tuning MCMC methods often do not exist and therefore they are often tuned in qualitative ways. QRS gives more to the user in this way. If the user has a fixed computation target, the user can get some idea of the quality of their samples within some budget. The same cannot be said for MCMC methods. This is a desirable property in a method intended to be used by people who may not be MCMC whizzes.

As well, the method is very simple and near trivial to implement which will greatly add to the impact of this method. Given a computation budget or a divergence target, the single parameter appears simple to tune and could even be done automatically (have you thought of this?).

Another nice property of the approach is that it can be easily applied on top of any new advances in learning the proposal distributions. If development is made in this area, QRS could easily be dropped-in on top to further improve results.

Weaknesses:

The main weakness in this work is in the empirical evaluation. The method is mainly compared with sampling only from the original proposal distribution. In this comparison, the method (clearly) appears favorable. As well, there is a comparison with Independent Metropolis-Hastings (IMH) in the appendix where there appears to be some benefit but it is not as drastic.

I believe a comparison with localized MCMC methods should also be provided. There has been recent development in this space and it would be useful to understand how well this approach works in the context of these recent developments. The same evaluations could be used as in Figure (b) in the appendix. You could seed the chains with your proposal distribution and then apply some MCMC algorithm for a certain number of steps. You could compare with any number of MCMC samplers such as random-walk metropolis, Gibbs sampling, locally-balanced proposals, gibbs-with-gradients, discontinuous HMC, discontinuous SGLD, etc (I am not asking you to compare with all of these approaches, just one or two maybe).

The x% acceptance rate QRS requires 1/x samples per accept. Since each sample requires L model evaluations (where L is sequence length) this requires L/x model evaluations in total. Metropolis requires 2 evals, gibbs-with-gradients requires 4 (2 evals + 2 gradients), so a runtime-fair comparison could easily be obtained with many of these local MCMC approaches.

I want to reiterate that I enjoyed this work and would like to see it accepted, but I feel it is lacking context in its current presentation. The authors claim a major upside of QRS is due to a number of deficiencies in these methods, so I think a comparison with the most recent methods in this area is warranted. I am happy to facilitate a discussion about this and if you feel this request is unreasonable, I am happy to discuss.

Besides this, I am also somewhat concerned with the importance sampling estimators of TVD/KL/accept-rate. Importance sampling maybe unbiased but it can have high variance. Could you also provide some information on the variance of the estimates used in your experiments?

---------Post discussion period------------
I thank the authors for responding to my comments. I appreciate the new experiments on localized MCMC. It appears that the proposed approach gives improved performance for a similar compute budget. While I would love to have seen a GWG comparison, I understand the overhead that it would require and am fine that it was not included. Unlike the other reviewers, I do not see the "lack of novelty" as an issue of this work. It should be favored when simpler ideas, applied in novel ways or to new problems, lead to good results on important problems. Based on the authors response, I will raise my score slightly in favor of acceptance.


**Summary Of The Paper:**

This paper focuses on the challenge of sampling from unnormalized probability distributions (or Energy-Based Models) -- typically those found in the context of NLP when dealing with the task of controlled generation (post-hoc conditioning of a pre-trained unconditional model). Previous approaches for this have been based on MCMC or have sought to finetune the unconditional model to more closely match the desired conditional model (usually through policy-gradient methods).

This work proposes a different method of sampling from such distributions, motivated by the accessibility of powerful generative models. Previous approaches have shown how to train a tractable generative model (which can be sampled from exactly) to approximate some known, unnormalized distribution. This work proposes to use these models as a proposal distribution for a rejection-sampling-like algorithm which will arrive at samples that are distributed more closely to the desired distribution.

Exact rejection sampling requires the user to know some upper bound M on the likelihood ratio between the desired distribution p(x) and the chosen proposal distribution q(x) [M > p(x)/q(x) \forall x]. In practice, this M may not be known, or may not exist. The proposed method, Quasi-Rejection Sampling (QRS), alleviates the need for this bound and replaces it with a tunable parameter \beta. This broadens the settings where we can apply this method compared to standard rejection sampling.

The authors demonstrate that as \beta \leftarrow \infty then TVD/KL between the output samples and target distribution goes to zero. This comes at the cost of increased rejection-rate which decreases the efficiency of the sampler. To make this approach more usable, the authors also provide importance sampling estimators for the TVD/KL which, along with the acceptance rate, can be used to tune \beta and provide estimates of sample quality.

The authors demonstrate the effectiveness of their approach in a number of controlled text generation settings and demonstrate that it can lead to improved samples over prior works (at the cost of additional computation).


**Summary Of The Review:**

The proposed approach, QRS, is a novel extension of rejection-sampling which can be applied in settings where standard rejection-sampling cannot. The approach has a number of benefits over alternative sampling approaches such as the ability to estimate useful metrics for tuning. The approach can bootstrap on top of further development for learning proposal distributions to target unnormalized distributions. While the paper is well written and the method appears powerful and novel, the paper has issues with its evaluation. QRS is not compared with sequential sampling approaches. This leaves out much context and ignores much recent work on discrete sampling. The paper would be greatly improved with a comparison to some of these approaches.

---

> ### Author Response · Authors · 2021-11-24
> **RE: Official Review of Paper2817 by Reviewer d6gm**
>
> Thank you for your questions and constructive feedback. We would like to point you to our main rebuttal post above (in 3 parts) and appendices H, I and J in the revised paper. In particular, regarding your comments and questions we:
> - Provide **additional experiments on localized MCMC methods** in appendix H.
> - Share some more thoughts on alternative localized approaches in appendix I.
> - Provide **variance estimates of our importance sampling estimators** in appendix J.
>
> Please do also see a summary of the results in the main rebuttal post. We are looking forward to hearing your thoughts on this and would be happy to answer any further questions during this final stage of discussion.

---

### Official Review · Reviewer_RQGP · 2021-11-03

**Correctness:** 3
**Technical Novelty And Significance:** 1
**Empirical Novelty And Significance:** Not applicable
**Recommendation:** 1
**Confidence:** 5

**Main Review:**

## Weaknesses

1 - The authors propose a relaxation of rejection sampling which is using an arbitrary parameter $\beta$ instead of the true upper bound of the ratio $\frac{p}{q}$ when the latter cannot be computed. The reviewer fails to understand why the authors did not directly use Importance sampling in the first place.

2- In algorithm 1, the reviewer fails to see a difference between QRS and RS, and will change their opinion if the authors can point out a value of `u` for which QRS and RS will behave differently.

3 - Uninteresting Section 2.2:
    - Equation 1 needs a parenthesis to avoid confusion
    - Equation 3 is pretty much obvious from the definition of TVD (Lemma 2.19 Aldous and Fill https://www.stat.berkeley.edu/users/aldous/RWG/book.html)
    - Equation 4 is obvious since using the apropriate upper bound gives you rejection sampling which is a perfect sampling algorithm.

4 - In the abstract the authors require the proposal distribution to upper bound the target everywhere which is not true as the authors themselves clarify in the text.

5 - While Equation 9 and 10 are great in that they can be used to compute TVD and KL between the true and QRS distributions, there are multiple issues which are neither stated as assumptions nor addressed appropriately, namely:
    - They're not unbiased estimators since $Z$ is not known and needs to be estimated, this point is not explicitly stated.
    - It is assumed that the normalizing constant of $q$ is known, which is not always the case.
    - They rely on importance sampling, which begs question 1.


**Summary Of The Paper:**

The authors propose a method called Quasi Rejection Sampling (`QRS`) which is a relaxation of Rejetion Sampling (`RS`).
The only difference between `RS` and `QRS` is that the authors use a paramater $\beta$ instead of the upper bound $B = \sup \frac{p}{q}$ used in traditional Rejection sampling, where $q$ is the proposal and $p$ is the target distribution.

**Summary Of The Review:**

The reviewer strongly recommends rejection since the paper lacks both significance and novelty.

---

> ### Author Response · Authors · 2021-11-29
> **RE: Official Review of Paper2817 by Reviewer RQGP**
>
> *“The reviewer fails to understand why the authors did not directly use Importance sampling in the first place.”*
>
> (Reviewer's addition on 23 Nov 2021) *“One could use importance subsampling. If you are using an arbitrary upper bound, why not sample  samples from the proposal distribution and sub-sample proportional to the importance weights. This procedure generates samples according to the target distribution as the number of samples goes to infinity.”*
>
> (Reviewer's addition on 23 Nov 2021) *“[Added later] Also please read point 1 and 5 in my evaluation together. You propose a rejection sampling algorithm, which you use importance sampling to tune. Since you're using importance sampling within your procedure anyways, it there a reason why you can't use it for directly sampling as well?”*
>
>
> As we have stated in our submission, we are interested in **obtaining samples** from an arbitrary distribution from which we can score. Importance Sampling (IS, https://artowen.su.domains/mc/Ch-var-is.pdf) is an algorithm which is used to **estimate expectations**, something we use extensively in our work. We do not make use of importance sampling within our sampler, however, but only for estimating diagnostics (KL, TVD, etc.). Note that **importance ratios** $P(x)/q(x)$ are part of many sampling algorithms (e.g. RS, Metropolis-Hastings, etc.). The “importance subsampling” procedure mentioned in the 23 November addition by Reviewer RQGP is akin to the “Sampling Importance Resampling” algorithm (SIR, Rubin, 1987), one of many such algorithms. A core problem with most generic samplers, however, is that even though they often reach perfect sampling in the limit, they do not provide good diagnostics for assessing the quality of **approximate** samples. For QRS we show that we can score under the sampler and hence estimate (using IS) divergence to the target distribution allowing for an informed trade-off between efficiency and sampling quality.
>
> *“In algorithm 1, the reviewer fails to see a difference between QRS and RS, and will change their opinion if the authors can point out a value of u for which QRS and RS will behave differently.”*
>
> The crucial difference between QRS and RS is that in QRS the bound $\beta q(x) \geq P(x)$ **is not enforced**. This means that the importance ratio $P(x)/(\beta q(x))$ may exceed 1, in which case we always accept the sample. For clarity, in order to keep the interpretation of $r_x$ to be the acceptance probability of the sample, we cut it off at 1. This is not essential for the practical implementation of the algorithm (as the uniform draw is always below 1). The crucial difference of QRS from RS does *not* lie in this line of the algorithm, but rather in the fact that *we do not enforce the bound*, while providing explicit diagnostics for the divergence when the bound is violated.
>
> *“In the abstract the authors require the proposal distribution to upper bound the target everywhere which is not true as the authors themselves clarify in the text.”*
>
> (Reviewer's addition on 23 Nov 2021) *“This is what you say: "For instance, rejection sampling can provide exact samples but is often difficult or impossible to apply due to the need to find a proposal distribution that upperbounds the target distribution everywhere." Even in the current version of the paper.”*
>
> The quoted sentence from the abstract refers to **rejection sampling**, in which case a proposal distribution and value of $\beta$ must be found such that they provide a strict upper-bound on the target distribution. Again, not enforcing this upper-bound is what distinguishes QRS from RS.
>
> *“Empirical Novelty And Significance: Not applicable”.*
>
> Already in the initial submission, we provided a number of experiments that (1) clearly showed the difference between RS and QRS (Two Poissons experiment), and (2) made clear the trade-off permitted by QRS between efficiency and approximation quality, in different experimental conditions exploiting different original forms of global proposals (coming from Distributional Policy Gradients, In-Context Learning, Back-Translation), and also (in the Appendix) an experimental comparison between QRS and IMH. Therefore, we believe that there is substantial empirical content in our work.

---

### Official Review · Reviewer_rWrS · 2021-11-08

**Correctness:** 2
**Technical Novelty And Significance:** 2
**Empirical Novelty And Significance:** 2
**Recommendation:** 5
**Confidence:** 3

**Main Review:**

The proposed method is very simple and easy to use in practice. However I have the below concerns.

1.	The main concern I have is how efficient the proposed method is given a usable sampling quality. From the results in the experiment section, to reach a reasonable TVD (e.g. < 0.001), the acceptance rate has to be very low which means QRS needs many steps to produce one sample. Besides, the hyperparameter beta and the proposal distribution need to be carefully chosen.

2.	Given the above concern, I wonder how the proposed method compared to MCMC methods. Given a similar number of steps, the MCMC method might also be able to mix and produce usable samples. This comparison is important and it is currently missing. For example, the authors could consider comparing with [Oops I Took A Gradient: Scalable Sampling for Discrete Distributions, ICML 2021] where they also tested on EBM, and the independent Metropolis-Hastings as mentioned in the paper.

3.	Algorithm 2 with automatically tuned beta is interesting since beta is an important hyperparameter and may require a lot of tuning to make QRS work. It will be better to show some experimental results of Algorithm 2. Right now it is not clear to me what the distribution it samples from.  Since beta changes along with the training, the target distribution also changes.



**Summary Of The Paper:**

This paper introduces Quasi Rejection Sampling to balance sampling accuracy and efficiency for energy-based models. Specifically, the authors introduce a hyperparameter beta to the standard rejection sampling to truncate the target distribution. By doing so, a bound on the importance ratio is not needed but the target distribution is no longer preserved.


**Summary Of The Review:**

In summary I think the methodology is simple and practical. It could have potential usage on many applications. But the empirical demonstration is not convincing, especially missing the comparison with MCMC methods.

---

> ### Author Response · Authors · 2021-11-24
> **RE: Official Review of Paper2817 by Reviewer rWrS**
>
> Thank you for your questions and constructive feedback. We would like to point you to our main rebuttal post above (in 3 parts) and appendices H, I and J in the revised paper. In particular, regarding your comments and questions we:
> - Address your concerns regarding the efficiency of QRS in the main rebuttal post.
> - Provide **additional experiments comparing with various MCMC methods** in appendices H and I.
> - Provide (and point to existing) **experiments using algorithm 2**, along with some clarifications on the algorithm in the main rebuttal post.
>
> For more details please refer to the main rebuttal post. We are looking forward to hearing your thoughts on this and would be happy to answer any further questions during this final stage of discussion.

---

### Author Response · Authors · 2021-11-23
**Author Response 1/3**

**We thank reviewers rWrS (R1) and d6gm (R3) for their thoughtful and constructive feedback, and will try to answer their comments and questions below.**

Unfortunately, **RQGP (R2)** considers our submission worthless (assessment: 1), is maximally self-confident (confidence: 5), but does not understand even the most basic aspects of our work:

- *“The reviewer fails to understand why the authors did not use importance sampling in the first place”* : this despite the obvious and clearly explained fact that we want to do *sampling*, not estimation of expectations (as *importance sampling* does).
- *“In algorithm 1, the reviewer fails to see a difference between QRS and RS”* : this remark indicates a total lack of attention to this central and very easy aspect of our submission.
- *“Uninteresting Section 2.2”* : very negative comment, without reasonable justification (the reviewer does not pay attention to basic facts about our claims).
- *“In the abstract the authors require the proposal distribution to upper-bound the target everywhere …”* : in complete contradiction to what we say in the abstract.

This review is below technical and ethical standards of reviewing and we do not address it anymore here.

&nbsp;

## Comparison to MCMC techniques

**rWrS (R1)** and **d6gm (R3)** both stress the lack of comparison with MCMC techniques for discrete spaces relying on local proposals (“localized MCMC”), in particular at an experimental level.  Let’s first note (**R1**) that concerning MCMC with _global_ proposals, such as Independent Metropolis-Hastings (IMH), we did already provide (Appendix E) both a theoretical and an experimental comparison with QRS, with some observations also valid for localized MCMC to which we will come back below.

On the other hand, concerning **localized MCMC** specifically, we do agree that our consideration of this was inadequate. We use section H in the appendix (see the updated submission) to provide an extended discussion of this topic along with new experiments. If the paper is accepted, we will move some of that discussion to the main text, but due to space (we are not allowed an additional page in the main text) and time constraints, we do not do it now.

In these new experiments, we focus on discrete Random Walk Metropolis-Hastings (RWMH) for text generation, conceptually close to (Miao et al. 2018) with some ingredients from (Goyal et al. 2021). We did consider GwG (Gibbs-with-Gradients (Grathwohl et al. 2021)) and other related gradient-based proposals, but felt that attempting to apply such techniques for text generation tasks such as those considered in our main experiments would be very ambitious and worth a paper on its own, see below.

These experiments address tasks considered in section 3.2 (Generation with Distributional Control), both pointwise and distributional constraints, with several versions of RWMH using different levels of customization of the local proposal, along with a comparison of the “standard” (non i.i.d) version of RWMH that follows a single random walk with burn-in and thinning with a “reset” (i.i.d) version of RWMH. We compare results of these samplers with QRS and IMH at similar “efficiency” levels and overall find that:

- In terms of constraints satisfaction, perplexity and Self-BLEU, local proposals (RWMH) tend to work much worse than global proposals, even with some customization of the local proposal. It is possible that with a high degree of customization and/or expensive proposals locally informed by the EBM (typically requiring a lot of EBM evaluations around the current point, a problem that GwG tries to alleviate, see below and Appendix I), this situation could be improved, but we did not go this route.
- Under the standard (non-reset) regime, MCMC methods show a large degree of autocorrelation and repetition of sequences.
- The two i.i.d, approaches IMH-reset and QRS, based on a global proposal, are on par in terms of moments, self-BLEU and perplexity, but only QRS can explicitly estimate the actual distance to the target EBM in terms of TVD or KL.

---

> ### Author Response · Authors · 2021-11-23
> **Author Response 3/3**
>
> ## Other points raised by reviewers
>
> ### Efficiency of QRS
>
> **R1** inquires about the efficiency of the QRS sampler. First and foremost, let’s note that efficiency is a function of the quality of the proposal distribution and its "closeness" to the target distribution, and not necessarily an inherent property of the algorithm. In our experiments on controlled text generation we found that we could achieve near-zero TVD around an acceptance rate of 10^-3 (1 in 1000 samples). This could indeed be too inefficient for one's application, and leaves the practitioner with two options: 1) try to find a better proposal distribution, 2) sacrifice some TVD for higher efficiency. The first option might not be trivial to do at all, while in some applications 2) might be a very reasonable option and a TVD of 0.1 or higher is even acceptable (note that in our CTG experiments, we already get very decent moment matching for a TVD around 0.3 at 10^-1 acceptance rate). The benefit we have over other methods is that we can actually estimate the TVD and KL values to the target distribution, at any acceptance rate, allowing for an informed tradeoff for the practitioner. Other sampling methods, like MCMC, do not allow for explicit estimates of such quantities, and might very well need similar acceptance rates or worse for getting a TVD < 0.001, but crucially we do not know!
>
> ### On choosing beta and algorithm 2
> **R3** inquires about the possibility of automatically tuning $\beta$, while **R1** requires more details on Algorithm 2. Algorithm 2 in Appendix D automatically estimates $\beta$ to a desired level of efficiency and then returns a set of samples as if using Algorithm 1 with the resulting value of $\beta$. Experimental results using this algorithm are shown in Appendix G where we present the empirical moments of a set of samples at different acceptance rate levels (marked with red stars). Our new results for QRS in Appendix H are also generated using Algorithm 2 in order to meet the target acceptance rate. Following the reviewers' feedback, we have made minor modifications to the algorithm to make its operation clearer.
>
> ### On variance of Importance Sampling
> (**R3**) We provide some variance estimates of our importance sampling estimates of AR, TVD and KL in Appendix J. We find our estimates to be accurate within reasonable variance.
>
> &nbsp;&nbsp;
>
> **We thank R1 and R3 for their feedback.** We believe their suggestions will help us improve the paper in its final version. Furthermore, we hope that the discussion above and the new sections in the appendix, which include new experimental results, have addressed their concerns. If they find our answers convincing, we would be grateful if they would update their evaluations.
>
> **To conclude,** we would like to stress the potential impact of sampling techniques that can exploit global proposals, such as ours, in contrast to those that make use of local proposals. Indeed, the results of the new experiments suggested by the reviewers corroborate the superiority of QRS but also of IMH with respect to RWMH. Perhaps even more importantly for practical applications, we note that while deep learning methods have made high-quality global proposals readily available, thus allowing a straightforward application of sampling techniques that can exploit them, local proposals still require a labour-intensive process of adapting the proposal to the desired EBM to make the approach feasible. Nonetheless, in the space of sampling techniques that can exploit global proposals, QRS strikes us as a much better alternative than IMH for a very simple reason: It provides convergence diagnostics that allow us to assess the quality of the approximation in a precise way without relying on proxy metrics such as perplexity, constraint satisfaction, Self-BLEU, etc., which are domain-specific and do not give us the full picture. For all of these reasons, we believe that our work fills in an important gap in MC sampling in discrete spaces and thus, can very positively impact the field.

---

> ### Author Response · Authors · 2021-11-23
> **Author Response 2/3**
>
> **Other observations relative to a “comparison of local MCMC methods with the proposed approach”**:
>
> - **Caveat on Evaluation**. The goal of a sampler $\omega$ for an EBM $P(x)$ is to get samples distributionally close to $p(x)$, the normalized distribution associated with $P$. Unfortunately, as is well-known, estimating divergences, such as KL or TVD, between $p$ and an MCMC sampler $\omega$ is very difficult, if possible at all. In our MCMC experiments, we instead only report constraint satisfaction, perplexity, and Self-BLEU (Zhu et al, 2018). It should be remarked here that two samplers with total constraint satisfaction and with equal perplexities (“fluencies”) relative to some language model can have very different divergences from $p$, in particular because the first one can cover a small part of $p$ while the second one can be close to $p$ everywhere. In order to partly compensate for this we also assess the _diversity_ of the samples produced by $\omega$, in terms of Self-BLEU. However, this provides only an unreliable indication of the closeness of $\omega$ to $p$, and should be interpreted with care.
> Note that with an MCMC sampler $\omega$ it is typically _impossible_ to estimate the score $\omega(x)$ for a given $x$. This contrasts strongly with the case of the QRS sampler $p_\beta$, which is also a scorer that can estimate $p_\beta(x)$ for any $x$, allowing equations (9) and (10) to be exploited and to estimate the TVD and KL divergences from $p$.
> _Put a bit more bluntly, we are not aware of any published evaluation measure of MCMC samplers for text generation that is able to characterise their ability to approach the underlying goal of approximating the distribution $p$, with literature often stopping at obtaining a good fluency and constraint satisfaction (for example (Miao et al. 2018) , (Goyal et al. 2021)). We believe that it is a large methodological advantage of QRS that it is able to address the underlying goal directly_.
> -**Genericity of Global Proposals vs. Local Proposals**: We already observed in the submission that recent advances in NN approximations to EBMs have increased the availability of reasonable global proposals for QRS, and of course this is also true for IMH. Some of these advances (e.g. DPG fine-tuning) are of a generic nature: they automatically produce decent approximations to the EBM -- seen as a blackbox -- with little intervention. Such a genericity seems much more difficult to obtain with local proposals in MCMC (see Appendix H), unless one introduces local proposals that are not pre-filtered based on knowledge of the EBM, leading to a very unfocused and inefficient process. We believe that aspect to be an important practical advantage of techniques exploiting global proposals, namely QRS and IMH (where we additionally argue that QRS is preferable to IMH due to general issues with MCMC).
> - **Gibbs-with-Gradients and other gradient-based techniques for discrete spaces**: Both **R1** and **R3** mention  “Gibbs with Gradients” (GwG, aka “Oops I took a Gradient … “ (Grathwohl et al. 2021)), a recent promising technique for importing the effective gradient techniques of continuous EBMs (as in vision) to discrete EBMs. We note, however, that (Grathwohl et al 2021) do not experiment with textual data, and we believe that while applying their technique to some text generation EBMs similar to those of our current experiments would be possible (please refer to Appendix I for our sketch of an approach towards this), this would be an ambitious research project on its own, beyond the scope of the current paper, as would be experiments with related gradient-based techniques such as “continuous relaxations” (see Appendix I for a description of the differences with GwG).

---

> ### Comment · Reviewer_RQGP · 2021-11-23
> **Authors resort to ad-hominem instead of responding to questions.**
>
> “The reviewer fails to understand why the authors did not use importance sampling in the first place” : this despite the obvious and clearly explained fact that we want to do sampling, not estimation of expectations (as importance sampling does).
>  - One could use importance subsampling. If you are using an arbitrary upper bound, why not sample $n$ samples from the proposal distribution and sub-sample proportional to the importance weights. This procedure generates samples according to the target distribution as the number of samples goes to infinity.
> - [Added later] Also please read point 1 and 5 in my evaluation together. You propose a rejection sampling algorithm, which you use importance sampling to tune. Since you're using importance sampling within your procedure anyways, it there a reason why you can't use it for directly sampling as well?
>
> “In algorithm 1, the reviewer fails to see a difference between QRS and RS” : this remark indicates a total lack of attention to this central and very easy aspect of our submission.
> - I asked you a direct question which you didn't answer.
>
> “Uninteresting Section 2.2” : very negative comment, without reasonable justification (the reviewer does not pay attention to basic facts about our claims).
> - I justified why each insight in Section 2.2 was uninteresting. I went through your proofs and independently verified if what you say makes sense, which is evidenced by my claim regarding the TVD result.
>
> “In the abstract the authors require the proposal distribution to upper-bound the target everywhere …” : in complete contradiction to what we say in the abstract.
> - This is what you say: "For instance, rejection sampling can provide exact samples but is often difficult
> or impossible to apply due to the need to find a proposal distribution that upperbounds the target distribution everywhere." Even in the current version of the paper.
>
> This review is below technical and ethical standards of reviewing and we do not address it anymore here.
> - You decided to resort to an ad-hominem attack instead of answering specific questions that I asked in order to understand the contribution.

---

### Decision · Program_Chairs · 2022-01-20

**Decision:**

Reject

**Comment:**

This paper proposes a new approximate sampling approach called Quasi Rejection Sampling (QRS) to exploit global proposal distributions without requiring to know a bound on the associated importance ratio, and providing a trade-off between the approximation quality of the
sampler and its efficiency. QRS is demonstrated on EBM-based text generation tasks. The reviews acknowledge the simplicity of the approach which when combined with advances in learning proposal distributions opens up many potential applications. At the same time, the reviews indicate that more work could be done to make the empirical demonstrations more compelling, with a more thorough coverage of comparisons with MCMC and other alternatives. The authors are encouraged to revise their submission and clarify significance and novelty.